# A Cross-Moment Approach for Causal Effect Estimation

**Yaroslav Kivva**
School of Computer and Communication Sciences
EPFL, Lausanne, Switzerland
`yaroslav.kivva@epfl.ch`

**Saber Salehkaleybar**
School of Computer and Communication Sciences
EPFL, Lausanne, Switzerland
`saber.salehkaleybar@epfl.ch`

**Negar Kiyavash**
College of Management of Technology
EPFL, Lausanne, Switzerland
`negar.kiyavash@epfl.ch`

## Abstract

We consider the problem of estimating the causal effect of a treatment on an outcome in linear structural causal models (SCM) with latent confounders when we have access to a single proxy variable. Several methods (such as difference-in-difference (DiD) estimator or negative outcome control) have been proposed in this setting in the literature. However, these approaches require either restrictive assumptions on the data generating model or having access to at least two proxy variables. We propose a method to estimate the causal effect using cross moments between the treatment, the outcome, and the proxy variable. In particular, we show that the causal effect can be identified with simple arithmetic operations on the cross moments if the latent confounder in linear SCM is non-Gaussian.In this setting, DiD estimator provides an unbiased estimate only in the special case where the latent confounder has exactly the same direct causal effects on the outcomes in the pre-treatment and post-treatment phases. This translates to the common trend assumption in DiD, which we effectively relax. Additionally, we provide an impossibility result that shows the causal effect cannot be identified if the observational distribution over the treatment, the outcome, and the proxy is jointly Gaussian. Our experiments on both synthetic and real-world datasets showcase the effectiveness of the proposed approach in estimating the causal effect.

## 1 Introduction

Estimating the effect of a treatment (or an action) on an outcome is an important problem in many fields such as healthcare [SJS17], social sciences [Gan10], and economics [IR]. Randomized control trials are the gold standard to estimate causal effects. However, in many applications, performing randomized experiments are too costly or even infeasible, say due to ethical or legal concerns. Thus, estimating the causal effect from merely observational studies is one of the main topics of interest in causal inference. This problem has been studied extensively in two main frameworks, potential outcome (PO) framework [Rub74] and structural causal model (SCM) framework [Pea09]. The main

37th Conference on Neural Information Processing Systems (NeurIPS 2023).

quantity of interest in PO framework is the individual-based response variable, i.e., the value of outcome for a specific individual in the population considering a particular value for the treatment. In SCM framework, a set of structural causal assignments are defined to describe the data generation mechanism among a set of variables. This set of assignments is often represented by a directed acyclic graph (DAG) to show the causal relationships among the variables in the model. It can be shown that the two frameworks are logically equivalent in the sense that any theorem in one can be translated to the other [PJS17].

Difference-in-Difference (DiD) [L$^+$11] is one of the most frequently used non-experimental methods to estimate the effect of a treatment by comparing the average of outcome before and after applying the treatment in a treatment and control group. In fact, 26 of 100 most cited papers published by the American Economic Review used some variant of DiD or two-way fixed effect (an extension to multi-group and multi-time slots) to estimate the causal effect [DCD22]. DiD is an estimation process in PO framework for the setting where we have access to a population partitioned into control and treatment groups. The goal is to estimate the effect of treatment $D$ on outcome $Y$ where $D$ is equal to one if a treatment is given to an individual and zero otherwise. It is also assumed that the value of the outcome is observed just before giving any treatment (this pre-treatment value is denoted by $Z$) and it can be seen as a proxy variable for latent common causes of $D$ and $Y$. DiD method computes the causal effect by subtracting the difference of average outcome in two groups before applying treatment (i.e., $\mathbb{E}[Z|D=1] - \mathbb{E}[Z|D=0]$) from the one after the treatment (i.e., $\mathbb{E}[Y|D=1] - \mathbb{E}[Y|D=0]$). It can be shown the output of DiD is an unbiased estimate of the causal effect under some assumptions such as the parallel/common trend assumption which states that the outcome of the treatment group would have followed the same trend as the control group in the absence of the treatment (see (2) for the exact definition).

Although the initial setting of DiD is in PO framework, its counterpart in the SCM framework was considered in the negative outcome control approach [SRC$^+$16]. A negative outcome variable is a type of proxy variable that is not causally affected by the treatment. The causal graph in this approach is represented in Figure 1 where the unmeasured common cause of $D$ and $Y$ is represented by a latent variable $U$ and $D$ is not a cause of proxy variable $Z$. The causal effect of $D$ on $Y$ cannot be identified from the observational distribution over

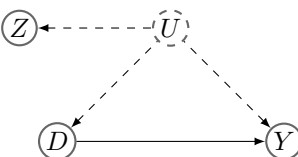

Figure 1: The suggested causal graph in SCM framework for the approaches in DiD and negative outcome control.

$(D, Y, Z)$ because of the common confounder $U$. However, imposing further assumptions on the SCM, the causal effect of $D$ on $Y$ can become identified. Such assumptions include monotonicity [SRC$^+$16], knowledge of the conditional probability $P(Z|U)$ [KP14], or having at least two proxy variables [KP14, MGTT18, TYC$^+$20, CPS$^+$20], all of which may not hold in practice (see related work in Section 4 for a more detailed discussion). Recently, [SGKZ20] considered linear SCMs with non-Gaussian exogenous noise [1] and proposed a method that can identify the causal effect for the causal graph in Figure 1 from the observational distribution over $(D, Y, Z)$. The proposed method is based on solving an over-complete independent component analysis (OICA) [HKO01]. However given the landscape of the optimization problem, in practice, OICA can get stuck in bad local minima and return wrong results [DGZT19].

In this paper, we consider the setup in causal graph in Figure 1 in linear SCM where we have access to a proxy variable $Z$. We propose a "Cross-Moment" algorithm that estimates the causal effect using cross moments between the treatment, the outcome, and the proxy variable. Our main contributions are as follows:

- We show that the causal effect can be identified correctly from the observational distribution if there exists $n \in \mathbb{N}$ such that for the latent confounder $U$, we have: $\mathbb{E}[U^n] \neq (n-1)\mathbb{E}[U^{n-2}]\mathbb{E}[U^2]$ (Theorem 1). Under additional mild assumption (Assumption 3), this condition implies that our proposed method can recover the causal effect when $U$ is non-Gaussian. Additionally, when the observational distribution is jointly Gaussian, we prove that it is impossible to identify the causal effect uniquely (Theorem 3).

[1]More precisely, at most one of the exogenous noises in the system can be Gaussian.

- Unlike previous work [SGKZ20, AHZ21, YGN$^+$22] which requires solving an OICA problem, the proposed approach only performs simple arithmetic operations on cross moments. Therefore, it does not suffer from the drawbacks of OICA such as getting stuck in bad local optima.
- We show that DiD estimator in general provides a biased estimate of the causal effect over the data generated by the linear SCM consistent with the causal graph in Figure 1 unless the latent confounder has exactly the same values of direct causal effects on the outcomes in the pre-treatment and post-treatment phases. Our proposed method does not require such a strong restriction.

The structure of the paper is as follows. In Section 2, we define the notation and provide some background on DiD estimator. In Section 3, we describe Cross-Moment algorithm and show that it recovers the true causal effect under mild assumptions on the distribution of the latent confounder. We also show that DiD estimator is in general biased if the data generative model follows a linear SCM. In Section 4, we review the related work. In Section 5, we evaluate the proposed algorithm experimentally and show its superior performance compared to the state of the art. Finally, we conclude the paper in Section 6.

## 2   Preliminaries and Notations

Throughout the paper, we denote random variables by capital letters and their realizations by small letters e.g., $X$ and $x$, respectively. Bold capital letters are used to specify a set of random variables and their realizations are denoted by small capital letters (e.g., $\mathbf{X}$ and $\mathbf{x}$).

A SCM $\mathcal{M}$ over a set of random variables $\mathbf{V}$ is defined by a set of assignments $\{X := f_X^{\mathcal{M}}(Pa_{\mathcal{G}}(X), \epsilon_X)\}_{X \in \mathbf{V}}$, where $\epsilon_X$ is the exogenous noise corresponding to $X$ and $Pa_{\mathcal{G}}(X) \subseteq \mathbf{V}$. It is assumed that the exogenous noises are mutually independent. Let us denote by $\mathbf{O}$ and $\mathbf{U}$, the set of observed and unobserved variables in $\mathbf{V}$, respectively. Note that $\mathbf{V} = \mathbf{O} \cup \mathbf{U}$ and $\mathbf{O} \cap \mathbf{U} = \emptyset$.

The set of assignments in SCM $\mathcal{M}$ is commonly represented by a DAG. Let $\mathcal{G} = (\mathbf{V}, \mathbf{E})$ be a DAG with the set of vertices $\mathbf{V}$ and set of edges $\mathbf{E}$. For ease of notation, we use the notation of $\mathbf{V}$ for the set of vertices in the graph. We also use the term "vertex" and "random variable" interchangeably. Each vertex in the graph represents some random variable and each direct edge shows a direct causal relationship between a pair of random variables. In particular, we say that $X$ is a parent of $Y$ or, equivalently, $Y$ is a child of $X$ if $(X, Y) \in \mathbf{E}$. We define $Pa_{\mathcal{G}}(X)$ as a set of all parents of $X$ in graph $\mathcal{G}$.

### 2.1   Difference-in-Difference (DiD)

Difference-in-difference (DiD) was proposed in the PO framework in order to estimate the causal effect from observational studies under some assumptions. In this framework, the population under study is divided into control and treatment groups and only individuals in the treatment group receive the treatment. In particular, the treatment variable $D$ represents treatment assignment which is equal to 1 if the treatment was given and 0 otherwise. Let $Y(0)$ and $Y(1)$ be two random variables representing the outcome under treatment value $D = 0$ and $D = 1$, respectively. Denote the value of the outcome right before administering the treatment by $Z$ and assume it is measurable. Our goal is to obtain the average causal effect in the treatment group $\mathbb{E}[Y(1) - Y(0)|D = 1]$. DiD estimate of the average causal effect equals:

$$(\mathbb{E}[Y|D = 1] - \mathbb{E}[Y|D = 0]) - (\mathbb{E}[Z|D = 1] - \mathbb{E}[Z|D = 0]). \tag{1}$$

This quantity is an unbiased estimate of the average causal effect as long as the following assumptions hold.

- Stable Unit Treatment Value Assumption (SUTVA):
$$Y = DY(1) + (1 - D)Y(0).$$

- Common trend assumption:
$$\mathbb{E}[Y(0) - Z(0)|D = 1] = \mathbb{E}[Y(0) - Z(0)|D = 0]. \tag{2}$$

SUTVA states that the potential outcome for each individual is not related to the treatment value of the other individuals. The common trend assumption states that there would be the same "trend" in both groups in the absence of treatment which allows us to subtract group-specific means of the outcome in estimating the average causal effect in (1).

# 3 Methodology: Cross-Moment Algorithm

In this section, we propose Cross-Moment algorithm to estimate the causal effect of treatment $D$ on outcome $Y$. Throughout this section, we consider linear SCMs, i.e., each random variable in SCM $\mathcal{M}$ is a linear combination of its parents and its corresponding exogenous noise. More precisely, the linear assignments in $\mathcal{M}$ for the causal graph in Figure 2 are:

$$
\begin{aligned}
U &:= \epsilon_u, \\
Z &:= \alpha_z U + \epsilon_z = \alpha_z \epsilon_u + \epsilon_z, \\
D &:= \alpha_d U + \epsilon_d = \alpha_d \epsilon_u + \epsilon_d, \\
Y &:= \beta D + \gamma U + \epsilon_y = (\alpha_d \beta + \gamma)\epsilon_u + \beta \epsilon_d + \epsilon_y,
\end{aligned}
\tag{3}
$$

Without loss of generality, we assume that $\epsilon_u$, $\epsilon_y$, $\epsilon_z$, $\epsilon_d$ are arbitrary random variables with zero mean. This assumption can always be achieved by centering the observational data.

Moreover, we assume that the only observed random variables are given by $\mathbf{O} = \{D, Y, Z\}$. Our goal is to identify $\beta$ (the causal effect of $D$ on $Y$) from the distribution over the observed variables $\mathbf{O}$. We consider SCMs that satisfy the following assumption.

**Assumption 1.** In the linear SCM given by (3), $\alpha_z \neq 0$ and $\mathrm{Var}(\epsilon_d) > 0$.

Assumption 1 is necessary for identifying the causal effect. In particular, if $\alpha_z = 0$, the directed edge from $U$ to $Z$ is removed and the causal effect cannot be identified even if all the

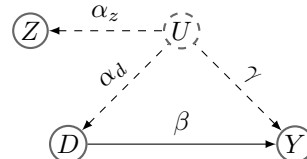

Figure 2: The considered causal graph with linear assignments in the SCM framework.

exogenous noises are non-Gaussian as shown in [SGKZ20]. Moreover, if $\mathrm{Var}(\epsilon_d) = 0$, then $\epsilon_d$ is zero almost surely (as we assumed that all the exogenous noises are mean zero). In this case, we can construct another SCM $\mathcal{M}'$ which encodes the same observational distribution as our original SCM but results in a different value of the causal effect of $D$ on $Y$ compared to the original SCM. More specifically, in this SCM, we delete the directed edge from $U$ to $Y$ and change the structural assignment of $Y$ to $Y := (\beta + \gamma/\alpha_d)D + \epsilon_y$. Hence, the assumption $\mathrm{Var}(\epsilon_d) > 0$ is necessary for the unique identification of the causal effect.

Under Assumption 1, it can be shown that:

$$
\beta = \frac{\mathrm{Cov}(D, Y) - \frac{\alpha_d}{\alpha_z}\mathrm{Cov}(Y, Z)}{\mathrm{Var}(D) - \frac{\alpha_d}{\alpha_z}\mathrm{Cov}(D, Z)},
\tag{4}
$$

where $\mathrm{Cov}(A, B)$ denotes the covariance of random variables $A$ and $B$ and $\mathrm{Var}(A)$ is the variance of $A$. $\beta$ is identifiable as long as the ratio $\alpha_d/\alpha_z$ is known as we can obtain $\mathrm{Cov}(D, Y), \mathrm{Cov}(Y, Z), \mathrm{Var}(D)$, and $\mathrm{Cov}(D, Z)$ from the observational distribution. In the sequel, we will show how this ratio can be learnt as long as $\epsilon_u$ has bounded moments.

**Assumption 2.** For all $n \in \mathbb{N}$, assume that: $\mathbb{E}[\epsilon_u^n] < \infty$.

When the bounded moment assumption of 2 holds, the following theorem provides an approach for recovering $\alpha_d/\alpha_z$.

**Theorem 1.** *For variables $Z$ and $D$ as defined in* (3)*, under Assumptions 2, $\frac{\alpha_d}{\alpha_z}$ can be determined uniquely if $\exists n \in \mathbb{N}$ such that:*

$$
\mathbb{E}\left[\hat{\epsilon}_u^n\right] \neq (n-1)\mathbb{E}\left[\hat{\epsilon}_u^{n-2}\right]\mathbb{E}\left[\hat{\epsilon}_u^2\right],
\tag{5}
$$

*where $\hat{\epsilon}_u = \sqrt{\alpha_d \alpha_z}\epsilon_u$.*

The detailed proof of the Theorem 1 is provided in the Appendix A.

It is interesting to see for what families of distributions, the condition in Theorem 1 is satisfied. Assume (5) is not satisfied. Recall that from the definition of SCM (3), $\mathbb{E}[\hat{\epsilon}_u] = 0$ and $\mathbb{E}[(\hat{\epsilon}_u)^2] = \mathbb{E}[DZ]$. These in a combination with $\mathbb{E}\left[\hat{\epsilon}_u^n\right] = (n-1)\mathbb{E}\left[\hat{\epsilon}_u^{n-2}\right]\mathbb{E}\left[\hat{\epsilon}_u^2\right]$ for any $n \in \mathbb{N}$ determine

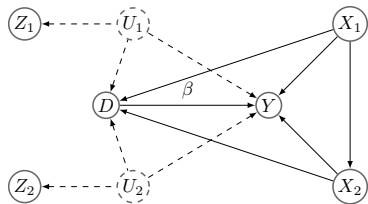

Figure 3: Example of causal graph extended with two observed covariates, two latent confounders, and corresponding proxy variables.

uniquely all the moments of $\hat{\epsilon}_u$. More specifically, recursively solving for $\mathbb{E}\left[\hat{\epsilon}_u^n\right]$ we have $\mathbb{E}\left[\hat{\epsilon}_u^n\right] = (n-1)!!\mathbb{E}[(\hat{\epsilon}_u)^2]$ for even $n \geq 1$ and $\mathbb{E}\left[\hat{\epsilon}_u^n\right] = 0$ for odd $n \geq 1$ where $n!!$ denotes double factorial. Double factorial notation $n!!$ denotes the product of all numbers from 1 to $n$ with the same parity as $n$. Specifically the moments of Gaussian distribution satisfy the aforementioned moment equation. Therefore when $\epsilon_u$ is Gaussian, we cannot identify the causal effect. Under some mild technical assumption on $\epsilon_u$ (see Assumption 3 in the following), we can prove that the moments of $\epsilon_u$ *uniquely* determine its distribution. As a result, as long as $\epsilon_u$ is non-Gaussian, we can identify the causal effect.

**Assumption 3.** We assume that there exists some $s > 0$ such that the power series $\sum_k \mathbb{E}[\epsilon_u^k]r^k/k!$ converges for any $0 < r < s$.

**Corollary 1.** *Under Assumptions 1, 2 and 3, the causal effect $\beta$ can be recovered uniquely as long as $\epsilon_u$ is not Gaussian.*

In [SGKZ20], it was shown that $\beta$ can be recovered as long as *all* exogenous noises are non-Gaussian. Therefore, our result relaxes the restrictions on the model in [SGKZ20] by allowing $\epsilon_D, \epsilon_Y, \epsilon_Z$ to be Gaussian.

Based on Theorem 1, we present Cross-Moment algorithm in Algorithm 1 that computes coefficient $\beta$ from the distribution over the observed variables $Z, D, Y$. Algorithm 1 is comprised of two functions *GetRatio* and *GetBeta*. In the proof of Theorem 1, we show that $|\alpha_d/\alpha_z| = (\text{num/den})^{1/(n-2)}$ for the smallest $n$ such that den $\neq 0$ where num and den are defined in lines 6 and 7 of function *GetRatio*, respectively. Moreover, $\mathbb{E}[DZ]$ has the same sign as $\alpha_d/\alpha_z$ and we can recover the sign of the ratio $\alpha_d/\alpha_z$ from $\mathbb{E}[DZ]$. Thus, in lines 8-10, for the smallest $n$ such that den $\neq 0$, we obtain the ratio $\alpha_d/\alpha_z$ and then use it in function *GetBeta* to recover $\beta$.

For ease of presentation, we presented the Cross-Moment algorithm for the specific causal graph in Figure 2 with only one proxy. However, the Cross-Moment algorithm can be utilized in a more general setting with additional covariates and latent confounders such as in the graph depicted in Figure 3. This generalization is stated in the following theorem.

**Theorem 2.** *Suppose that the linear SEM of* (3) *with the graph in Figure 2 is extended with observed covariates* **X***, non-Gaussian latent confounders* **U** *and proxy variables* **Z** *such that*

- *none of the observed covariate is a descendant of any latent variable;*

- *no latent confounder $U \in$ **U** of variables $D$ and $Y$ is an ancestor of any other latent confounder;*

- *for each latent confounder $U \in$ **U** there exists a unique proxy variable $Z \in$ **Z** which is not an ancestor of $Y$;*

- *each latent confounder and its unique proxy variable satisfy the Assumptions 1, 2 and 3.*

*Then the causal effect $\beta$ from $D$ to $Y$ can be computed uniquely from the observational distribution.*

The proof of the theorem appears in Appendix A. The main idea of the proof is to reduce the problem to a set of sub-problems that can be solved with Cross-Moment algorithm. As mentioned earlier, an example of the causal graph satisfying the conditions of the theorem is depicted in Figure 3.

---

**Algorithm 1** Cross-Moment algorithm

---

1: **Function** *GetBeta*$(D, Z, Y)$
2: ratio := *GetRatio*$(D, Z)$
3: $\beta := (\mathbb{E}[DY] - \text{ratio} \cdot \mathbb{E}[YZ])/(\mathbb{E}[D^2] - \text{ratio} \cdot \mathbb{E}[DZ])$
4: **Return** $\beta$

---

1: **Function** *GetRatio*$(D, Z)$
2: findRatio := **False**
3: $n := 2$
4: **while** findRatio $\neq$ **True do**
5:    $n := n + 1$
6:    num := $\mathbb{E}[D^{n-1}Z] - (n-1)\mathbb{E}[D^{n-2}]\mathbb{E}[DZ]$
7:    den := $\mathbb{E}[Z^{n-1}D] - (n-1)\mathbb{E}[Z^{n-2}]\mathbb{E}[DZ]$
8:    **if** den $\neq 0$ **then**
9:       ratio := **sign**$(\mathbb{E}[DZ]) \left|(\frac{\text{num}}{\text{den}})^{1/(n-2)}\right|$
10:       findRatio := **True**
11:    **end if**
12: **end while**
13: **Return:** ratio

---

### 3.1 Impossibility Result

In the previous sections, we showed that the causal effect $\beta$ can be identified if the distribution of latent confounder is non-Gaussian. Herein, we show that no algorithm can learn $\beta$ uniquely if the observed variables are jointly Gaussian in any linear SCM defined by (3) satisfying the following assumption.

**Assumption 4.** In the linear SCM defined by (3), $\alpha_d \neq 0$, $\gamma \neq 0$ and $\text{Var}(\epsilon_z) > 0$.

**Theorem 3.** *Suppose that the observed variables in linear SCM defined by* (3) *are jointly Gaussian. Under Assumptions 1, 2 and 4, the total causal effect $\beta$ cannot be identified uniquely.*

The proof of the Theorem 3 appears in the Appendix A. The key idea in the proof is to show that there exist two linear SCMs that encode the same observational distribution and are consistent with the causal graph in Figure 2 but the causal effect of $D$ on $Y$ has two different values in these two models.

Note that it is known that the causal structure is not identifiable in a linear SCM with Gaussian exogenous noises [PJS17]. Our impossibility result here is different from the non-identifiability result in linear Gaussian models. Specifically, in the linear Gaussian models, the goal is to recover all the coefficients in the linear SCM from the observational distribution. In our setting, we have additional knowledge of the exact DAG (restriction on the form of the linear SCM in (3)), and the goal is to identify a specific coefficient (i.e., $\beta$) from the linear SCM. Therefore, we have more constraints on the model and need to infer less information about it. Still, we show that the target coefficient $\beta$ cannot be determined in the causal graph in Figure 2 if the observed variables are jointly Gaussian.

### 3.2 Bias in DiD Estimator

Suppose that the data is generated from a linear SCM consistent with the causal graph in Figure 2. We show that DiD estimator is biased except when the latent variable $U$ has the exact same direct causal effect on $Z$ that it has on $Y$, i.e., $\alpha_Z = \gamma$. Our Cross-Moment algorithm identifies the true causal effect without any such restrictive assumption on the coefficients of the linear SCM.

DiD estimator is given by the following linear regression [L$^+$11]:

$$\hat{Y} = \hat{\beta}_1 T + \hat{\beta}_2 D + \hat{\beta} DT, \tag{6}$$

where $\hat{\beta}_1$, $\hat{\beta}_2$, and $\hat{\beta}$ are the regression coefficients and $T$ is a binary variable that equals zero for the pre-treatment phase and equals one otherwise. In the pre-treatment phase, $Z$ (i.e., the outcome before the treatment) is predicted as $\hat{\beta}_2 D$ and in the post-treatment phase, $Y$ (the outcome after giving treatment to the treatment group) is predicted accordingly as $\hat{\beta}_1 + (\hat{\beta} + \hat{\beta}_2)D$. In order to obtain

the regression coefficients, the expectation of squared residuals over the population is minimized as follows (see Appendix C for the derivations of the following minimization problem and subsequent equations in this section):

$$\min_{\hat{\beta}_1, \hat{\beta}_2, \hat{\beta}} \mathbb{E}[(Z - \hat{\beta}_2 D)^2] + \mathbb{E}[(Y - \hat{\beta}_1 - (\hat{\beta} + \hat{\beta}_2)D)^2].$$

This results in the following regression coefficients:

$$\hat{\beta}_1 = 0, \quad \hat{\beta}_2 = \frac{\mathbb{E}[ZD]}{\mathbb{E}[D^2]}, \quad \hat{\beta} = \frac{\mathbb{E}[YD] - \mathbb{E}[ZD]}{\mathbb{E}[D^2]}.$$

DiD estimator returns $\hat{\beta}$ in the above equation as the estimation of causal effect which is equal to:

$$\hat{\beta} = \beta + \frac{\alpha_d(\gamma - \alpha_z)\mathbb{E}[U^2]}{\mathbb{E}[D^2]}. \tag{7}$$

Thus, $\hat{\beta}$ is an unbiased estimate of $\beta$ only when $\gamma = \alpha_z$. In other words, latent variable $U$ should have the same direct causal effect on $Z$ and $Y$. This is akin to the so-called common trend assumption which says that the average natural drift (here, the effect of $U$) is assumed to be the same across both the control and treatment groups. This result is consistent with the findings in [RSBP23], which studied a similar phenomenon in the DiD setting. In summary, whenever the common trend assumption is violated, the DiD estimator is biased.

## 4  Related work

In the past few years, there has been a growing interest in the literature to exploit proxy variables to de-bias the effect of latent confounders. A special type of such proxy variable is the so-called negative outcome which is a variable known not to be causally affected by the treatment [LTC10]. For instance, the variable $Z$ in Figure 1 may be considered as negative outcome. In fact, [SRC+16] interpreted DiD as a negative outcome control approach and proposed a method inspired by change-in-change [AI06] to identify the causal effect under the assumption that $Y(0)$ and $Z$ are monotonic increasing functions of latent confounders and some observed covariates.

[KP14] considered three settings in causal inference with proxy variables: 1- There exists only one proxy variable such as $Z$ as a negative outcome. In this case, for discrete finite variables $Z$ and $U$, they showed that the causal effect can be identified if $\Pr(Z|U)$ is known from some external studies such as pilot studies. 2- Two proxy variables, for instance $Z$ and $W$ are considered where $U$, $Z$, and $W$ are all discrete finite variables and $Z$ does not have a directed path to $D$ or $Y$. It has been shown that the causal effect is identifiable under some assumptions on the conditional probabilities of $\Pr(Y|D, U)$ and $\Pr(Z, W|X)$. In the setting, it is not necessary to know $\Pr(Z|U)$ but two proxy variables are required to identify the causal effect. 3- In linear SCM, [KP14] showed that $\beta$ (the average causal effect of $D$ on $Y$) can be recovered using two proxy variables. Later, [MGTT18] also considered a setting with two proxy variables $Z$ and $W$. Unlike the second setting in [KP14], here, $Z$ and $W$ can be parents of $D$ and $Y$, respectively. For the discrete finite variables, they showed that the causal effect can be identified if the matrix $P(W|Z, D = d)$ is invertible. Moreover, they provided the counterpart of this condition for continuous variables. [SMNTT20] extended the identification result in [MGTT18], with a weaker set of assumptions. Still, they required two proxy variables to identify the causal effect. Based on the results in [MGTT18], [TYC+20] introduced a proximal causal inference in PO framework. Later, [CPS+20] provided an alternative proximal identification result to that of [MGTT18], again when two proxy variables were avaialble. More recently, [SLZ+23] considered the PO framework under linearity assumptions for treatment and post-treatment phase. In this setting, the authors showedthe identifiability of causal effect if in the pre-treatment phase, the latent confounder jointly with observed outcome follows a multivariate Gaussian distribution, and in the treatment phase, the exogenous noise of outcome variable is non-Gaussian.

In linear SCMs, to the best of our knowledge, the methods that can identify the causal effect with only one proxy variable in Figure 2 are based on solving an OICA problem. In particular, [SGKZ20] considered linear SCM with non-Gaussian exogenous noises in the presence of latent variables. They showed that under some structural conditions, the causal effects among observed variables can be identified and the causal graph in Figure 2 satisfies such structural conditions. [YGN+22] extended the results in [SGKZ20]. They showed that the causal structure (direction) can always be identified, and the causal effect can be identified up to equivalence classes depending on the graphical

conditions (the causal graph in Figure 2 is still uniquely identifiable). However, both proposed methods [SGKZ20, YGN$^+$22] require solving an OICA, and existing algorithms for solving such a problem might get stuck in bad local optima. Recently, [AHZ21] provided two graphical conditions for the same setting in [SGKZ20] which are necessary for the identification of the causal structure. These conditions are closely related to the sparsity of the causal graphs. For the causal graph in Figure 2, the method proposed in [AHZ21] for estimating the causal effect is the same as the one in [SGKZ20] and thus has the same drawback. Concurrent to our submission, [CHC$^+$23] considered the problem of causal discovery for the linear non-Gaussian models. Under the specific assumptions on the true causal graph, they proposed an algorithm for learning the causal structure as well as the causal coefficients using high-order cumulants.

In PO framework, the setting of having just a pre-treatment phase and a post-treatment phase can be generalized to the case with multiple time slots in the panel data model [ABD$^+$21]. In this paper, we mainly focus on the setting with two groups and two time slots but one can also study the extensions of the current work for other settings in the panel data model described in the following. Consider two $N \times T$ matrices $\mathbf{Y}$ and $\mathbf{D}$ where $N$ is the number of individuals in the population and $T$ is the number of time slots. Assume that only the outcome for some individuals and time slots is observable. In particular: $Y_{it} = (1 - D_{it})Y_{it}(0) + D_{it}Y_{it}(1)$, where the realized outcome for individual $i$ at time slot $t$ is denoted by $Y_{it}(D_{it})$. DiD method has been proposed for the case $T = 2$, i.e., two time slots (pre-treatment and post-treatment phases). In the literature, other cases have been also studied for various assumptions on matrix $\mathbf{Y}$. For instance, in unconfounded case [RR83, IR], the number of individuals is much larger than the number of time slots ($N \gg T$), and the treatment is provided only at the last time slot. Another setting is that of synthetic control [AG03, ADH10, ADH15, DI16] where $T \gg N$. In this setting, there is a single treated individual (suppose individual $N$) and the goal is to estimate its missing potential outcomes for any $t \in [T_0, T]$ after administering the treatment at time $T_0$. The last setting considers $N \approx T$ and a two-way-fixed-effect (TWFE) regression model has been proposed to estimate the causal effect (see for a survey on TWFE in [DCd20]). It is noteworthy that TWFE estimator is equivalent to DiD estimator for two groups and two time slots.

## 5 Experiments

In this section, we first evaluate our algorithm on synthetic data and compare it to DiD estimator and as well as the related work in [KP14] which estimates the causal effect in linear SCMs with two proxy variables. Further, we apply our algorithm to a real dataset provided by [CK93]. The implementation of the algorithm and additional experimental results are provided in https://github.com/ykivva/Cross-Moments-Method.

### 5.1 Synthetic data

We generated samples according to the SCM in (3) and with all the exogenous noises distributed according to exponential distribution. Note that the distribution of $\epsilon_u$, i.e., the exponential distribution satisfies Assumptions 2 and 3. Therefore $\beta$ is identifiable according to the Corollary 1.

Given the observational data, we estimated the value of $\beta$ from the following four approaches:

1. Cross-Moment algorithm (proposed in this work).

2. DiD estimator of (6).

3. A simple linear regression model based on the following equation: $\hat{Y} = \alpha Z + \hat{\beta} D$.

4. Causal effect estimate for linear SCM with two proxy variables (proposed in [KP14]). In the experiments, we call this estimate "two-proxy" method.

It is noteworthy that we also evaluated the method in [SGKZ20] which uses OICA as a subroutine. Unfortunately, the performance was too poor to be included.

For each sample size, we sampled parameters $\alpha_z$, $\alpha_d$, $\beta$, $\gamma$ randomly and then generated the samples of $Z, D, Y$ accordingly to (3) (More details regarding the data generation mechanism can be found in Appendix B). We ran an experiment 10 times and reported the the average relative error for each value of sample size: $err = \mathbb{E}\left[\left|\frac{\beta - \hat{\beta}}{\beta}\right|\right]$.

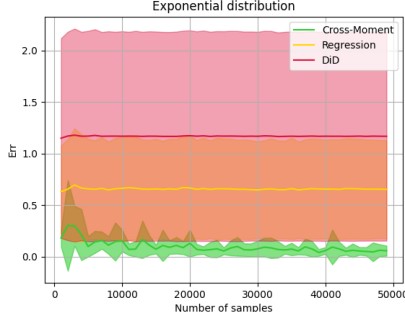
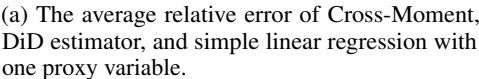
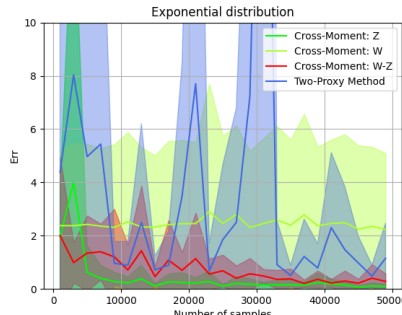

(a) The average relative error of Cross-Moment, DiD estimator, and simple linear regression with one proxy variable.

(b) The average relative error of three variants of Cross-Moment algorithm and the two-proxy method in [KP14] when we have access to two proxy variables.

Figure 4: The performance measure $err$ against the number of samples. Colored regions show the standard deviation of the $err$.

Figure 4a, depicts the performances of Cross-Moment algorithm, DiD estimator, and the simple linear regression when we have access to only one proxy variable. The colored region around each curve shows the empirical standard deviation of $|(\beta - \hat{\beta})/\beta|$. Cross-Moment algorithm outperforms the other two methods significantly. In fact, DiD estimator is biased if $\alpha_z \neq \gamma$ which occurs with measure one as $\alpha_z$ and $\gamma$ are generated randomly. Moreover, DiD estimate is no better than simple linear regression if $\gamma$ is not close to $\alpha_z$. In the literature, it has been noted that the parallel trend assumption (in linear SCM, this assumption is equivalent to the condition $\alpha_z = \gamma$) is violated if the scale of the proxy variable $Z$ and outcome variable $Y$ are different which can be the case in many practical applications [L+11].

We compared Cross-Moment with the two-proxy method in [KP14] when we have access to two proxy variables. In particular, we assumed that there is an additional proxy variable $W$ such that $W := \alpha_w U + \epsilon_w$. For Cross-Moment algorithm, we considered three versions: I - "Cross-Moment: Z", which estimates the causal effect by using only proxy variable $Z$ (we denote this estimate by $\beta_Z$), II - "Cross-Moment: W", which estimates $\beta$ from only proxy variable $W$ (which we denote the estimate by $\beta_W$), III - "Cross-Moment: W-Z", which estimates $\beta$ from aggregating the estimates of the methods I and II. In particular, "Cross-Moment: W-Z" uses bootstrapping method (Monte Carlo algorithm for case resampling [ET94]) to estimate the variances of estimates $\beta_Z$ and $\beta_W$, denoted by $\sigma^2_{\beta_Z}$ and $\sigma^2_{\beta_W}$, respectively. Subsequently, $\beta$ is estimated by combining two estimates $\beta_Z$ and $\beta_W$ with an inverse-variance weighting scheme [SHK11] where we give a higher weight to the estimate with the lower variance: $\frac{\sigma^2_{\beta_Z}}{\sigma^2_{\beta_Z} + \sigma^2_{\beta_W}} \beta_W + \frac{\sigma^2_{\beta_W}}{\sigma^2_{\beta_Z} + \sigma^2_{\beta_W}} \beta_Z$. When $\mathrm{Var}(\epsilon_w)/\mathrm{Var}(W)$ and $\mathrm{Var}(\epsilon_z)/\mathrm{Var}(Z)$ are small, the causal effect can be estimated with a low estimation error from either $Z$ or $W$ as they contain low noise versions of the latent confounder $U$. In our experiments, we considered the case where one of the proxy variables (herein, $W$) is too noisy but not the other one. Specifically, we choose $\mathrm{Var}(\epsilon_w)/\mathrm{Var}(\epsilon_u) = 10$ and $\mathrm{Var}(\epsilon_z)/\mathrm{Var}(\epsilon_u) = 0.1$. Figure 4b illustrates the performances of the three aforementioned variants of Cross-Moment algorithm and the two-proxy method in [KP14]. "Cross-Moment: Z" has the best performance since it uses $Z$ with less noise as the proxy variable. Moreover, "Cross-Moment: W-Z" has a comparable performance by combining the estimates of $\beta_Z$ and $\beta_W$. The two-proxy estimate does not exhibit robustness and has a large average relative error for various values of sample size.

## 5.2 Minimum Wage and Employment Dataset

We evaluate our method on the real data which contains information about fast-food stores (Burger King, Roy Rogers, and Wendy's stores) in New Jersey and Pennsylvania in 1992, and some details about them such as minimum wage, product prices, open hours, etc [CK93]. The goal of study was to estimate the effect of the increment in minimum wage in New Jersey from $ 4.25 to $ 5.05 per hour on the employment rate.

|  | TWFE | Cross-Moment |
|---|---|---|
| With $\mathbf{X}$ | 2.68 | 2.68 |
| Without $\mathbf{X}$ | 3.24 | 4.03 |

Table 1: Causal effect estimation of minimum wage on employment level in the real dataset in [CK93].

The data was collected by interviews in two waves, before and after the rise in the minimum wage. The information was gathered from 410 restaurants with similar average food prices, store hours, and employment levels. In this experiment, stores from Pennsylvania are treated as a control group and stores from New Jersey are considered as the treatment group. We define employment level $Y$ as $Y = Y_f + \frac{1}{2}Y_p$, where $Y_p$ is a number of employees working part-time and $Y_f$ is a number of employees working full-time.

First, we reproduced the results in [CK93]. We considered an extended version of TWFE model [CK93]:

$$\hat{Y} = \hat{\beta}_1 T + \mathbf{X}^T \hat{\alpha} + \hat{\beta}_2 D + \hat{\beta} DT,$$

where $\hat{Y}$ is the estimate of number of employees in the store, $T$ is a binary variable that equals $0$ prior to raising the minimum wage and equals to $1$ after the raise. $D$ is equal to $0$ if the store is in Pennsylvania and equal to $1$ if the store is in New Jersey. $\mathbf{X}$ is a vector that contains additional information such as the opening hours, product prices, etc. $\hat{\alpha}$ is also a vector of parameters corresponding to the vector $\mathbf{X}$. We dropped all the stores from the dataset that contain NaN values after which 246 restaurants were left. The estimate of $\beta$ computed by TWFE is given in the first row of Table 1. According to [CK93], the estimate of $\beta$ is equal to 2.76. The difference in estimation is due to the slight difference in the features of the vector $\mathbf{X}$, i.e., [CK93] added a few additional manually computed features to $\mathbf{X}$.

For the Cross-Moment algorithm, in order to incorporate the features $\mathbf{X}$ in estimating $\beta$, we first regressed $Y$ on $\mathbf{X}$ and then used $Y - \mathbf{X}\hat{\alpha}$ instead of $Y$ as the outcome. The result of applying Cross-Moment algorithm to this newly defined outcome is given in the first row of Table 1 and is very close to the estimate by TWFE.

Finally, we assumed that the additional information $\mathbf{X}$ gathered during the interview is not available. Then TWFE model for the employment level takes the following form

$$\hat{Y} = \hat{\beta}_1 T + \hat{\beta}_2 D + \hat{\beta} DT.$$

We used the previously pre-processed dataset but dropped the columns corresponding to $\mathbf{X}$. Subsequently, we applied TWFE and Cross-Moment method to estimate $\beta$. The respective estimates appear in the second row of Table 1, which stipulate the rise in the minimum wage had a positive effect on the employment level.

## 6 Conclusion

We considered the problem of estimating the causal effect of a treatment on an outcome in the linear SCM where we have access to a proxy variable for the latent confounder of the treatment and the outcome. This problem has been studied in both PO framework (such as DiD estimator) and SCM framework (such as the negative outcome control approach). We proposed a method that uses cross moments between the treatment, the outcome, and the proxy variable and recovers the true causal effect if the latent confounder is non-Gaussian. We also showed that the causal effect cannot be identified if the joint distribution over the observed variable are Gaussian. Unlike previous work which requires solving an OICA problem, our performs simple arithmetic operations on the cross moments. We evaluated our proposed method on synthetic and real datasets. Our experimental results show the proposed algorithm has remarkable performance for synthetic data and provides consistent results with previous studies on the real dataset we tested on.

## Acknowledgments and Disclosure of Funding

This research was in part supported by the Swiss National Science Foundation under NCCR Automation, grant agreement 51NF40_180545 and Swiss SNF project 200021_204355/1.

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
