## A Technical Proofs

**Theorem 1.** *For the variables $Z$ and $D$ as defined in (3), under Assumptions 2, $\frac{\alpha_d}{\alpha_z}$ can be determined uniquely if $\exists n \in \mathbb{N}$ such that:*

$$\mathbb{E}\left[\hat{\epsilon}_u^n\right] \neq (n-1)\mathbb{E}\left[\hat{\epsilon}_u^{n-2}\right]\mathbb{E}\left[\hat{\epsilon}_u^2\right], \tag{8}$$

*where $\hat{\epsilon}_u = \sqrt{\alpha_d \alpha_z}\epsilon_u$.*

*Proof.* Let $\alpha := \sqrt{\frac{\alpha_d}{\alpha_z}}$. Please note that $\alpha$ can be a complex number. Then, $D$ and $Z$ in (3) can be rewritten as:

$$D = \alpha\hat{\epsilon}_u + \epsilon_d,$$

$$Z = \frac{1}{\alpha}\hat{\epsilon}_u + \epsilon_z.$$

We prove by induction that:

$$\mathbb{E}\left[\hat{\epsilon}_u^n\right] = (n-1)\mathbb{E}\left[\hat{\epsilon}_u^{n-2}\right]\mathbb{E}\left[\hat{\epsilon}_u^2\right], \tag{9}$$

holds for any $n \in \mathbb{N}$ or $\frac{\alpha_d}{\alpha_z}$ ($\alpha^2$ equivalently) can be identified uniquely from $D$ and $Z$.

*Base of induction.* It is easy to verify that for $n = 2$, (9) holds.

*Induction step.* Assume that $n \geq 2$ and (9) holds for all $k < n$. Then we prove either it also holds for $n$ or $\alpha_d/\alpha_z$ can be identified uniquely. We have

$$\mathbb{E}[D^{n-1}Z] = \mathbb{E}[(\alpha\hat{\epsilon}_u + \epsilon_d)^{n-1}\frac{1}{\alpha}\hat{\epsilon}_u] = \tag{10}$$

$$\mathbb{E}[\alpha^{n-2}\hat{\epsilon}_u^n + \binom{n-1}{1}\alpha^{n-3}\hat{\epsilon}_u^{n-1}\epsilon_d + \cdots + \binom{n-1}{n-2}\hat{\epsilon}_u^2\epsilon_d^{n-2}].$$

By the induction hypothesis, we know that for all $k < n$:

$$\mathbb{E}[\hat{\epsilon}_u^k] = (k-1)\mathbb{E}[\hat{\epsilon}_u^{k-2}]\mathbb{E}[\hat{\epsilon}_u^2].$$

Then,

$$\binom{n-1}{k}\alpha^{n-k-2}\mathbb{E}[\hat{\epsilon}_u^{n-k}]\mathbb{E}[\epsilon_d^k] = \binom{n-1}{k}\alpha^{n-k-2}\mathbb{E}[\hat{\epsilon}_u^{n-k-2}]\mathbb{E}[\hat{\epsilon}_u^2]\mathbb{E}[\epsilon_d^k](n-k-1).$$

Note that:

$$\binom{n-1}{k}(n-k-1) = \frac{(n-1)!(n-k-1)}{(k)!(n-k-1)!} = \frac{(n-2)!(n-1)}{(k)!(n-k-2)!} = \binom{n-2}{n-k-2}(n-1).$$

Therefore,

$$\binom{n-1}{k}\alpha^{n-k-2}\mathbb{E}[\hat{\epsilon}_u^{n-k}]\mathbb{E}[\epsilon_d^k] = (n-1)\mathbb{E}[\hat{\epsilon}_u^2]\binom{n-2}{n-k-2}\mathbb{E}[\hat{\epsilon}_u^{n-k-2}]\mathbb{E}[\epsilon_d^k]. \tag{11}$$

Substituting all the terms except the first one in (10) using (11), we have:

$$\mathbb{E}[D^{n-1}Z] = \mathbb{E}[\alpha^{n-2}\hat{\epsilon}_u^n] + (n-1)\mathbb{E}[\hat{\epsilon}_u^2]\mathbb{E}\left[\sum_{k=0}^{n-2}\binom{n-2}{k}(\alpha\hat{\epsilon}_u)^k\epsilon_d^{n-2-k}\right]$$

$$- (n-1)\alpha^{n-2}\mathbb{E}[\hat{\epsilon}_u^{n-2}]\mathbb{E}[\hat{\epsilon}_u^2] = \alpha^{n-2}\mathbb{E}[\hat{\epsilon}_u^n] + (n-1)\mathbb{E}[\hat{\epsilon}_u^2]\mathbb{E}[D^{n-2}] - (n-1)\alpha^{n-2}\mathbb{E}[\hat{\epsilon}_u^{n-2}]\mathbb{E}[\hat{\epsilon}_u^2]$$

Consequently,

$$\mathbb{E}[D^{n-1}Z] - (n-1)\mathbb{E}[\hat{\epsilon}_u^2]\mathbb{E}[D^{n-2}] = \alpha^{n-2}\left(\mathbb{E}[\hat{\epsilon}_u^n] - (n-1)\mathbb{E}[\hat{\epsilon}_u^{n-2}]\mathbb{E}[\hat{\epsilon}_u^2]\right). \tag{12}$$

Similarly, we can get

$$\mathbb{E}[Z^{n-1}D] - (n-1)\mathbb{E}[\hat{\epsilon}_u^2]\mathbb{E}[Z^{n-2}] = \frac{1}{\alpha^{n-2}}\left(\mathbb{E}[\hat{\epsilon}_u^n] - (n-1)\mathbb{E}[\hat{\epsilon}_u^{n-2}]\mathbb{E}[\hat{\epsilon}_u^2]\right). \tag{13}$$

Note that the right hand sides of (12) and (13) should be equal to zero. Otherwise one can divide (12) by (13) and get the value of $\alpha^{2n-4}$. This is because we can obtain $\mathbb{E}[Z^{n-1}D]$ and $\mathbb{E}[D^{n-2}]$ from the observational distribution. The other term in the expression, $\mathbb{E}[\hat{\epsilon}^2]$, can also be computed from the observation distribution as it equals $\mathbb{E}[DZ]$. To see this, note that

$$\mathbb{E}[DZ] = \mathbb{E}\left[\hat{\epsilon}_u^2 + \hat{\epsilon}_u(\alpha\epsilon_z + \frac{1}{\alpha}\epsilon_d) + \epsilon_d\epsilon_z\right] = \mathbb{E}\left[\hat{\epsilon}_u^2\right].$$

Therefore, we can identify $\alpha^2$ uniquely up to its sign since $\alpha^2$ is a real-valued number. Furthermore the sign of $\frac{\alpha_d}{\alpha_z}$ is the same as the sign of the $\alpha_z\alpha_d\mathbb{E}[\hat{\epsilon}_u^2]$ which is equal to the $\mathbb{E}[DZ]$. Thus, $\frac{\alpha_d}{\alpha_z}$ will be determined uniquely if (9) is not satisfied for $n$ and the proof is complete. $\qquad\square$

**Corollary 1.** *Under Assumptions 1, 2 and 3, the causal effect $\beta$ can be recovered uniquely if $\epsilon_u$ is non-Gaussian.*

*Proof.* Based on [Bro83][Chapter 30, Theorem 30.1], under Assumption 3, the condition $\mathbb{E}\left[\hat{\epsilon}_u^n\right] = (n-1)\mathbb{E}\left[\hat{\epsilon}_u^{n-2}\right]\mathbb{E}\left[\hat{\epsilon}_u^2\right]$ is satisfied for any $n \in \mathbb{N}$ if and only if $\hat{\epsilon}_u$ is Gaussian. Thus, based on Theorem 1, the causal effect $\beta$ is identified if $\epsilon_u$ is non-Gaussian. $\qquad\square$

**Theorem 2.** *Suppose that the linear SEM given by* (3) *with the causal graph in Figure 2 is extended with covariates $\mathbf{X}$, non-Gaussian latent confounders $\mathbf{U}$ and proxy varialbes $\mathbf{Z}$ such that*

- *none of the observed covariate is a descendant of any latent variable;*

- *no latent confounder $U \in \mathbf{U}$ of variables $D$ and $Y$ is an ancestor of any other latent confounder;*

- *for each latent confounder $U \in \mathbf{U}$ there exists a unique proxy variable $Z \in \mathbf{Z}$ which is not an ancestor of $Y$;*

- *each latent confounder and its unique proxy variable satisfy the Assumptions 1, 2 and 3.*

*Then the causal effect $\beta$ from $D$ to $Y$ can be computed uniquely from the observational distribution.*

*Proof.* Let $\mathbf{X} = \{X_1, X_2, \ldots, X_k\}$ be the covariates. Since any of them is not a descendant of a latent confounder, then at least one of them should have no parents. Without loss of generality, suppose that $X_1$ has no parents and therefore we can write:

$$X_1 = \epsilon_{x_1}.$$

Let us consider any descendant $X_i$ of $X_1$. Then we have:

$$X_i = a\epsilon_{x_1} + R_{x_i},$$

where $R_{x_i}$ is independent of $\epsilon_{x_1}$ and $a$ is some constant. Regressing $X_i$ on $X_1$ we can recover $a$. Moreover, we can consider $X_i' := X_i - aX_1$ instead of $X_i$, where $X_1$ has no causal effect on $X_i'$. By applying this operation for all descendants of $X_1$, we reduce the problem to the one, with the same conditions as in the theorem but with a fewer set of covariates, that is $\{X_2, \ldots, X_k\}$. Continuing this procedure, we can eliminate all the covariates one by one and get the model in which we only have latent confounders.

Suppose that $U_0, U_1, \ldots, U_t$ are latent confounders specified in the theorem and $Z_0, Z_1, \ldots, Z_t$ are their corresponding unique proxies. Then we can write:

$$
\begin{aligned}
U_i &= \epsilon_{u_i} \ i \in \{0, 1, \ldots, t\}, \\
Z_i &= \alpha_{z_i}\epsilon_{u_i} + \epsilon_{z_i} \ i \in \{0, 1, \ldots, t\}, \\
D &= \epsilon_d + \sum_{i=0}^{t} \alpha_{d,u_i}\epsilon_{u_i}, \\
Y &= \epsilon_y + \sum_{i=0}^{t} \alpha_{\gamma,u_i}U_i + \beta D = \epsilon_y + \beta\epsilon_d + \sum_{i=0}^{t} \alpha_{\gamma,u_i}\epsilon_{u_i} + \beta\sum_{i=0}^{t} \alpha_{d,u_i}\epsilon_{u_i}.
\end{aligned}
$$

Since the Assumptions 1, 2 and 3 for the variables $Z_i$, $U_i$ and $D$ hold for each $i \in \{0, 1, \ldots, t\}$ and $U_i$ is non-Gaussian, we can apply Theorem 1. Using the Theorem 1 for $Z_i$, $U_i$ and $D$, we recover $\frac{\alpha_{d,u_i}}{\alpha_{z_i}}$. Moreover, we know that the following set of equalities are satisfied:

$$\text{Cov}(D, Y) = \beta \mathbb{E}[\epsilon_d^2] + \sum_{i=0}^{t} \alpha_{d,u_i} \left( \alpha_{\gamma,u_i} + \beta \alpha_{d,u_i} \right) \mathbb{E}[\epsilon_{u_i}^2],$$

$$\sum_{i=0}^{t} \frac{\alpha_{d,u_i}}{\alpha_{z_i}} \text{Cov}(Y, Z_i) = \sum_{i=0}^{t} \alpha_{d,u_i} \left( \alpha_{\gamma,u_i} + \beta \alpha_{d,u_i} \right) \mathbb{E}[\epsilon_{u_i}^2],$$

$$\text{Var}(D) = \mathbb{E}[\epsilon_d^2] + \sum_{i=0}^{t} \alpha_{d,u_i}^2 \mathbb{E}[\epsilon_{u_i}^2],$$

$$\sum_{i=0}^{t} \frac{\alpha_{d,u_i}}{\alpha_{z_i}} \text{Cov}(D, Z_i) = \sum_{i=0}^{t} \alpha_{d,u_i}^2 \mathbb{E}[\epsilon_{u_i}^2].$$

Using above equalities, it can easily be seen that $\beta$ is identified from following equation given the ratios $\{\frac{\alpha_{d,u_i}}{\alpha_{z_i}}\}$:

$$\beta = \frac{\text{Cov}(D, Y) - \sum_{i=0}^{t} \frac{\alpha_{d,u_i}}{\alpha_{z_i}} \text{Cov}(Y, Z_i)}{\text{Var}(D) - \sum_{i=0}^{t} \frac{\alpha_{d,u_i}}{\alpha_{z_i}} \text{Cov}(D, Z_i)},$$

and the proof is complete.

$\square$

**Theorem 3.** *Suppose that the observed variables in linear SCM in* (3) *have jointly Gaussian distribution. Under Assumptions 1, 2 and 4, the total causal effect $\beta$ cannot be identified uniquely.*

*Proof.* Without loss of generality, we assume that $Z$, $D$ and $Y$ have zero mean and are generated by a model $\mathcal{M}_1$ as follows:

$$\mathcal{M}_1 :$$
$$U = \epsilon_u,$$
$$Z = \alpha_z U + \epsilon_z,$$
$$D = \alpha_d U + \epsilon_d,$$
$$Y = \beta D + \gamma U + \epsilon_y,$$

and $\alpha_d = 1$. Otherwise, instead of $U = \epsilon_u$, one can write $U = \alpha_d \epsilon_u$ and rescale other coefficients respectively. Further, we construct a model $\mathcal{M}_2$ as follows:

$$\mathcal{M}_2 :$$
$$U = \epsilon_u,$$
$$Z = \frac{1}{k} \alpha_z U + \epsilon_z',$$
$$D = kU + \epsilon_d',$$
$$Y = \beta' D + \gamma' U + \epsilon_y',$$

where $\beta \neq \beta'$ and all the exogenous noises are Gaussian with a mean equal to 0 such that:

$$\text{Var}(Z)^{\mathcal{M}_1} = \text{Var}(Z)^{\mathcal{M}_1}, \quad \text{Var}(D)^{\mathcal{M}_1} = \text{Var}(D)^{\mathcal{M}_2}, \quad \text{Var}(Y)^{\mathcal{M}_1} = \text{Var}(Y)^{\mathcal{M}_2},$$
$$\text{Cov}(Z, D)^{\mathcal{M}_1} = \text{Cov}(Z, D)^{\mathcal{M}_2}, \quad \text{Cov}(Z, Y)^{\mathcal{M}_1} = \text{Cov}(Z, Y)^{\mathcal{M}_2}, \quad \text{Cov}(D, Y)^{\mathcal{M}_1} = \text{Cov}(D, Y)^{\mathcal{M}_2}.$$
$$\tag{14}$$

Since in both cases $Z$, $D$ and $Y$ are jointly Gaussian then both model agree on the distribution of observed variables. The latter means that total causal effect $\beta$ is not identifiable since $\beta \neq \beta'$ and it is impossible to distinguish between them having only observations of $Z$, $D$ and $Y$.

More specifically, we define $k = 1 - \delta$, where $\delta$ some real number such that:

$$0 < \delta < 1 - \sqrt{\frac{\alpha_z^2 \mathrm{Var}(\epsilon_u)}{\alpha_z^2 \mathrm{Var}(\epsilon_u) + \mathrm{Var}(\epsilon_z)}}, \tag{15}$$

$$\frac{\mathrm{Var}(\epsilon_d)}{\mathrm{Var}(\epsilon_u)} \geq (1 - k^2). \tag{16}$$

Accordingly, we define random variables $\epsilon_z'$, $\epsilon_d'$, $\epsilon_y'$ as Gaussian random variables with mean zero having the variances as follows:

$$\mathrm{Var}(\epsilon_z') := \sigma_z' = \alpha_z^2 \mathrm{Var}(\epsilon_u) + \mathrm{Var}(\epsilon_z) - \frac{1}{k^2} \alpha_z^2 \mathrm{Var}(\epsilon_u)$$

$$\mathrm{Var}(\epsilon_d') := \sigma_d' = \mathrm{Var}(\epsilon_u) + \mathrm{Var}(\epsilon_d) - k^2 \mathrm{Var}(\epsilon_u) > \mathrm{Var}(\epsilon_d)$$

$$\mathrm{Var}(\epsilon_y') := \sigma_y' = (\beta + \gamma)^2 \mathrm{Var}(\epsilon_u) + \beta^2 \mathrm{Var}(\epsilon_d) + \mathrm{Var}(\epsilon_y) - (k\beta' + \gamma')^2 \mathrm{Var}(\epsilon_u) - \beta'^2 \sigma_d',$$

where

$$\beta' := \beta + \gamma \mathrm{Var}(\epsilon_u) \left( \frac{1 - k^2}{\sigma_d'} \right),$$

$$\gamma' := k\gamma \frac{\mathrm{Var}(\epsilon_d)}{\sigma_d'}.$$

Further we will show that $\sigma_z' > 0$, $\sigma_y' > 0$ and such that the conditions in (14) hold, which completes the proof.

1.  *Here we will prove that $\sigma_z' > 0$ and*

$$\mathrm{Var}(Z)^{\mathcal{M}_1} = \mathrm{Var}(Z)^{\mathcal{M}_2}, \quad \mathrm{Cov}(Z, D)^{\mathcal{M}_1} = \mathrm{Cov}(Z, D)^{\mathcal{M}_2}.$$

From the inequality (15), we have:

$$k > \sqrt{\frac{\alpha_z^2 \mathrm{Var}(\epsilon_u)}{\alpha_z^2 \mathrm{Var}(\epsilon_u) + \mathrm{Var}(\epsilon_z)}} \implies k^2 > \frac{\alpha_z^2 \mathrm{Var}(\epsilon_u)}{\alpha_z^2 \mathrm{Var}(\epsilon_u) + \mathrm{Var}(\epsilon_z)} \implies$$

$$\alpha_z^2 \mathrm{Var}(\epsilon_u) + \mathrm{Var}(\epsilon_z) > \frac{\alpha_z^2}{k^2} \mathrm{Var}(\epsilon_u) \implies \sigma_z' = \alpha_z^2 \mathrm{Var}(\epsilon_u) + \mathrm{Var}(\epsilon_z) - \frac{\alpha_z^2}{k^2} \mathrm{Var}(\epsilon_u) > 0.$$

By the definition,

$$\mathrm{Var}(Z)^{\mathcal{M}_2} = \frac{\alpha_z^2}{k^2} \mathrm{Var}(\epsilon_u) + \mathrm{Var}(\epsilon_z') = \frac{\alpha_z^2}{k^2} \mathrm{Var}(\epsilon_u) + \alpha_z^2 \mathrm{Var}(\epsilon_u) + \mathrm{Var}(\epsilon_z) - \frac{\alpha_z^2}{k^2} \mathrm{Var}(\epsilon_u) =$$

$$\alpha_z^2 \mathrm{Var}(\epsilon_u) + \mathrm{Var}(\epsilon_z) = \mathrm{Var}(Z)^{\mathcal{M}_1},$$

and

$$\mathrm{Cov}(Z, D)^{\mathcal{M}_1} = \alpha_z \mathrm{Var}(\epsilon_u) = \mathrm{Cov}(Z, D)^{\mathcal{M}_2}.$$

2.  *Here we will prove that $\mathrm{Var}(D)^{\mathcal{M}_1} = \mathrm{Var}(D)^{\mathcal{M}_2}$.*

By the definition,

$$\mathrm{Var}(D)^{\mathcal{M}_2} = k^2 \mathrm{Var}(\epsilon_u) + \mathrm{Var}(\epsilon_d') = k^2 \mathrm{Var}(\epsilon_u) + \mathrm{Var}(\epsilon_u) + \mathrm{Var}(\epsilon_d) - k^2 \mathrm{Var}(\epsilon_u) =$$
$$\mathrm{Var}(\epsilon_u) + \mathrm{Var}(\epsilon_d) = \mathrm{Var}(D)^{\mathcal{M}_1}. \tag{17}$$

3.  *Here we will prove that:*

$$\mathrm{Cov}(Z, Y)^{\mathcal{M}_1} = \mathrm{Cov}(Z, Y)^{\mathcal{M}_2}, \quad \mathrm{Cov}(D, Y)^{\mathcal{M}_1} = \mathrm{Cov}(D, Y)^{\mathcal{M}_2}.$$

By the definition,

$$\mathrm{Cov}(Z, Y)^{\mathcal{M}_2} = \frac{1}{k} \alpha_z (\beta' k + \gamma') \mathrm{Var}(\epsilon_u) = \alpha_z \left( \beta + \gamma \mathrm{Var}(\epsilon_u) \left( \frac{1 - k^2}{\sigma_d'} \right) + \gamma \frac{\mathrm{Var}(\epsilon_d)}{\sigma_d'} \right) \mathrm{Var}(\epsilon_u) =$$

$$\alpha_z \left( \beta + \gamma \frac{\mathrm{Var}(\epsilon_u) + \mathrm{Var}(\epsilon_d) - k^2 \mathrm{Var}(\epsilon_u)}{\sigma_d'} \right) \mathrm{Var}(\epsilon_u) = \alpha_z (\beta + \gamma) \mathrm{Var}(\epsilon_u) = \mathrm{Cov}(Z, Y)^{\mathcal{M}_1}$$

and

$$\text{Cov}(D,Y)^{\mathcal{M}_2} = \beta'\text{Var}(D) + k\gamma'\text{Var}(\epsilon_u) = \left(\beta + \gamma\text{Var}(\epsilon_u)\left(\frac{1-k^2}{\sigma_d'}\right)\right)\text{Var}(D) + k^2\gamma\frac{\text{Var}(\epsilon_d)}{\sigma_d'}\text{Var}(\epsilon_u) =$$

$$\beta\text{Var}(D) + \gamma\text{Var}(\epsilon_u)\left(\frac{1-k^2}{\sigma_d'}\right)(\text{Var}(\epsilon_u) + \text{Var}(\epsilon_d)) + k^2\gamma\frac{\text{Var}(\epsilon_d)}{\sigma_d'}\text{Var}(\epsilon_u) =$$

$$\beta\text{Var}(D) + \gamma\text{Var}(\epsilon_u)\frac{(1-k^2)\text{Var}(\epsilon_u) + \text{Var}(\epsilon_d)}{\sigma_d'} = \beta\text{Var}(D) + \gamma\text{Var}(\epsilon_u) = \text{Cov}(D,Y)^{\mathcal{M}_1}$$

**4.** *Here we will proof that $\sigma_y' \geq 0$ and $\text{Var}(Y)^{\mathcal{M}_1} = \text{Var}(Y)^{\mathcal{M}_2}$.*

To get inequality $\sigma_y' \geq 0$ it is enough to show that

$$(\beta + \gamma)^2\text{Var}(\epsilon_u) + \beta^2\text{Var}(\epsilon_d) \geq (k\beta' + \gamma')^2\text{Var}(\epsilon_u) + \beta'^2\sigma_d'.$$

Therefore

$$(\beta + \gamma)^2\text{Var}(\epsilon_u) + \beta^2\text{Var}(\epsilon_d) \geq (k\beta' + \gamma')^2\text{Var}(\epsilon_u) + \beta'^2\sigma_d' \iff$$

$$(\beta + \gamma)^2\text{Var}(\epsilon_u) + \beta^2\text{Var}(\epsilon_d) \geq k^2\left(\beta + \gamma\text{Var}(\epsilon_u)\left(\frac{1-k^2}{\sigma_d'}\right) + \gamma\frac{\text{Var}(\epsilon_d)}{\sigma_d'}\right)^2\text{Var}(\epsilon_u) + \beta'^2\sigma_d' \iff$$

$$(\beta + \gamma)^2\text{Var}(\epsilon_u) + \beta^2\text{Var}(\epsilon_d) \geq k^2(\beta + \gamma)^2 + \left(\beta + \gamma\text{Var}(\epsilon_u)\left(\frac{1-k^2}{\sigma_d'}\right)\right)^2\sigma_d' \iff$$

$$(1-k^2)(\beta + \gamma)^2\text{Var}(\epsilon_u) + \beta^2\text{Var}(\epsilon_d) \geq \beta^2\text{Var}(\epsilon_d') + 2\beta\gamma\text{Var}(\epsilon_u)(1-k^2) + (1-k^2)^2 C,$$

where $C = \gamma^2\frac{\text{Var}(\epsilon_u)^2}{\sigma_d'}$. From (17), we can get

$$(1-k^2)(\beta + \gamma)^2\text{Var}(\epsilon_u) + \beta^2\text{Var}(\epsilon_d) \geq \beta^2\text{Var}(\epsilon_d') + 2\beta\gamma\text{Var}(\epsilon_u)(1-k^2) + (1-k^2)^2 C \iff$$
$$(1-k^2)(\beta + \gamma)^2\text{Var}(\epsilon_u) \geq (1-k^2)\beta^2\text{Var}(\epsilon_u) + 2\beta\gamma\text{Var}(\epsilon_u)(1-k^2) + (1-k^2)^2 C \iff$$
$$\gamma^2\text{Var}(\epsilon_u) \geq (1-k^2)C.$$

From the inequality (16) we have

$$\gamma^2\text{Var}(\epsilon_u) \geq \gamma^2(1-k^2)\frac{\text{Var}(\epsilon_u)^2}{\text{Var}(\epsilon_d)} \geq \gamma^2(1-k^2)\frac{\text{Var}(\epsilon_u)^2}{\sigma_d'} = (1-k^2)C.$$

The last inequality follows from the fact that $\sigma_d' \geq \text{Var}(\epsilon_d)$ (the definition of $\sigma_d'$). Therefore in our construction for the second model, we have $\sigma_y' \geq 0$.

Finally,

$$\text{Var}(Y)^{\mathcal{M}_2} = (\beta'k + \gamma')^2\text{Var}(\epsilon_u) + \beta'^2\sigma_d' + \text{Var}(\epsilon_y') =$$
$$(\beta'k + \gamma')^2\text{Var}(\epsilon_u) + \beta'^2\sigma_d' +$$
$$(\beta + \gamma)^2\text{Var}(\epsilon_u) + \beta^2\text{Var}(\epsilon_d) + \text{Var}(\epsilon_y) - (k\beta' + \gamma')^2\text{Var}(\epsilon_u) - \beta'^2\sigma_d' =$$
$$(\beta + \gamma)^2\text{Var}(\epsilon_u) + \beta^2\text{Var}(\epsilon_d) + \text{Var}(\epsilon_y) = \text{Var}(Y)^{\mathcal{M}_1}.$$

The above claims show that the two models are indistinguishable from the observational distribution and they have different causal effects of $D$ on $Y$ and thus the proof is complete. $\square$

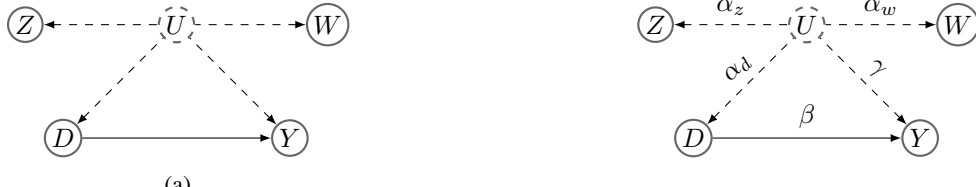

Figure 5: The considered causal graph for the experiments with linear assignments in the SCM framework.

## B  Experiments

### B.1  Synthetic data

For the experiments with synthetic data we assume that the samples are generated according to the following linear SCM:

$$
\begin{aligned}
U &:= \epsilon_u, \\
W &:= \alpha_w U + \epsilon_w, \\
Z &:= \alpha_z U + \epsilon_z, \\
D &:= \alpha_d U + \epsilon_d, \\
Y &:= \beta D + \gamma U + \epsilon_y.
\end{aligned} \tag{18}
$$

Given the observational data, we estimate the value of $\beta$ using the following methods and report the performances against the number of observed samples.

1. Cross-Moment algorithm proposed in this work.
2. DiD method according to (6).
3. A simple linear regression model based on the following equation: $\hat{Y} = \alpha Z + \hat{\beta} D$.
4. The "two-proxy" method in [KP14].
5. The algorithm proposed in [TYC$^+$20].
6. Experiment under misspecification.

For each value of sample size, we ran an experiment 10 times and reported the average relative error:

$$
err = \mathbb{E}\left[\left|\left|\frac{\beta - \hat{\beta}}{\beta}\right|\right|\right],
$$

and the standard deviation with the colored regions on the plots. Before each run, we randomly generate parameters $\alpha_d, \alpha_z, \alpha_w, \beta, \gamma$ as follows:

- $\alpha_d$ is randomly sampled from the interval $(-2, -0.2) \cup (0.2, 2)$,
- $\alpha_z, \beta, \gamma$ are randomly sampled such that the absolute value of the ratios between $\alpha_d$ and each of the variables $\alpha_z, \beta, \gamma$ are in the interval $(0.2, 2)$,
- We set $\alpha_w = \alpha_z$ to have a consistent setting for measuring the effect of noise in proxy variables on the Cross-Moment method and the two-proxy method proposed by [KP14].

In our experiments, the variances of $\epsilon_z, \epsilon_d, \epsilon_y$ are 10 time less than the variance of $\epsilon_u$. We also set the variance of $\epsilon_w$ to be 10 time bigger than the variance of $\epsilon_u$. Thus, proxy $W$ is much noisier compared with $Z$.

In the case of having two proxy variables $W, Z$, we combine the results of the Cross-Moment method applied for each proxy separately and called these overall procedure as "Cross-Moment: $W - Z$" method. More precisely "Cross-Moment: $W - Z$" method works as follows:

1. For $i$ in $[1 : t]$, randomly sample with replacement some portion of all observational data $(\mathbf{Z}, \mathbf{W}, \mathbf{D}, \mathbf{Y})$ that we denote by $(\mathbf{Z}_i, \mathbf{W}_i, \mathbf{D}_i, \mathbf{Y}_i)$.

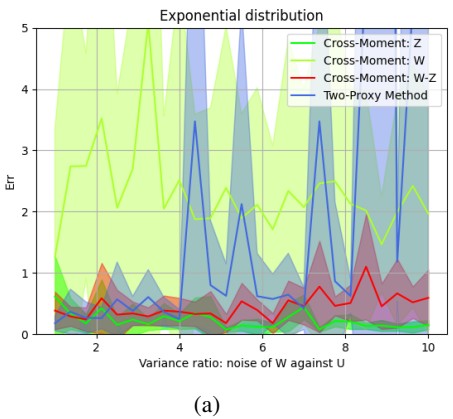 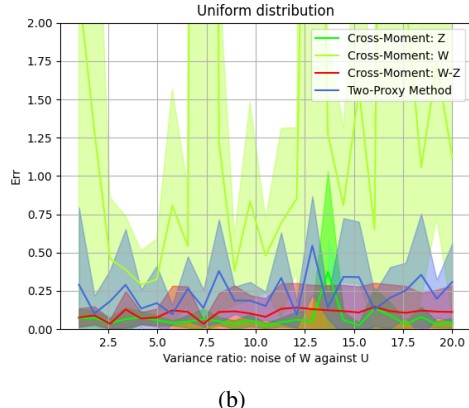

|(a)|(b)|

Figure 6: The performance measure $err$ against $\mathrm{Var}(\epsilon_w)/\mathrm{Var}(\epsilon_u)$.

2. Using Cross-Moment method over proxy variables $W$ and $Z$ separately, we estimate the causal effect $\beta$ from the data $(\mathbf{Z}_i, \mathbf{W}_i, \mathbf{D}_i, \mathbf{Y}_i)$. Let us denote these estimates as $\beta_W^{(i)}$ and $\beta_Z^{(i)}$ accordingly.

3. Having $\{\beta_W^{(i)}\}_{i=1}^t$ and $\{\beta_Z^{(i)}\}_{i=1}^t$, we approximate the variances of the estimates made from proxies $Z$ and $W$. We denote $\sigma_{\beta_W}^2$ and $\sigma_{\beta_Z}^2$, respectively

4. Compute the final estimation of $\beta$ as follows:

$$\hat{\beta} := \frac{\sigma_{\beta_W}^{-2}}{\sigma_{\beta_W}^{-2} + \sigma_{\beta_Z}^{-2}} \frac{\sum_{i=1}^t \beta_W^{(i)}}{t} + \frac{\sigma_{\beta_Z}^{-2}}{\sigma_{\beta_W}^{-2} + \sigma_{\beta_Z}^{-2}} \frac{\sum_{i=1}^t \beta_Z^{(i)}}{t}.$$

All the experiments were performed using 16 GB RAM and 12th Gen Intel(R) Core(TM) i7-12700H 2.30 GHz.

### B.1.1 Exponential distribution

Here we assume that all exogenous noises $\epsilon_u$, $\epsilon_z$, $\epsilon_w$, $\epsilon_d$, $\epsilon_y$ are from the class of exponential distributions. At the beginning of each run, we randomly choose the variance for the exogenous noise $\epsilon_u$ from the interval $(1, 10)$ and set all other exogenous noise distributions as we discussed in the previous section.

In addition to the experiments presented in the Section 5.1 on Exponetial distributions we also illustrate the performance of methods against the ratio between $\mathrm{Var}(\epsilon_w)$ to $\mathrm{Var}(\epsilon_u)$ in Figure 6a. To sum up we observe that two-proxy method suffers much more from the noise in the proxies and is much less stable than our "Cross-Moment W-Z" method.

In Figure 7, we compare "Cross-Moment" with the algorithm introduced by [TYC$^+$20] in a similar setting as we did for the "two-proxy" method. Here, we conclude again that our method is more stable against the noise in the given proxies.

### B.1.2 Uniform distribution

Here we assume that all the exogenous noises $\epsilon_u$, $\epsilon_z$, $\epsilon_w$, $\epsilon_d$, $\epsilon_y$ have uniform distribution. In this scenario, we considered the same setting as for the exponential distributions. At the beginning of each run, we consider the exogenous noise $\epsilon_u$ to be a uniform distribution on the interval $[-a, a]$, where $a$ is a random real number picked from the interval $(1, 10)$. In Figure 7a, we compare "Cross-Moment" method with the algorithm introduced by [TYC$^+$20].

Figure 8 illustrates the performance of the Cross-Moment and Two-Proxy methods with respect to the number of observed samples. Again, we observe that "Cross-Moment W-Z" method more stable than Two-Proxy method. Additionally, in Figure 6b, we show the dependence of the performance of the methods on the ratio between $\mathrm{Var}(\epsilon_w)/\mathrm{Var}(\epsilon_u)$. Although, for the uniform distribution two-proxy

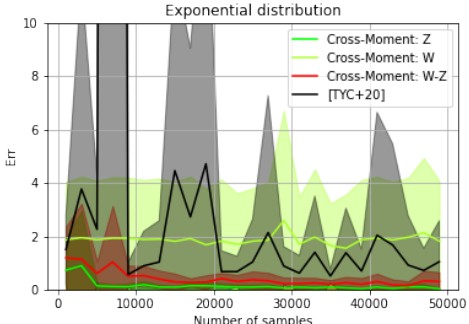
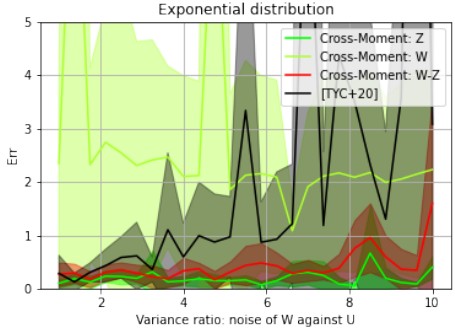

(a) The average relative error of three variants of Cross-Moment algorithm and the method in [TYC+20].

(b) The performance measure $err$ against $\mathrm{Var}(\epsilon_w)/\mathrm{Var}(\epsilon_u)$ for three variants of Cross-Moment algorithm and the method in [TYC+20].

Figure 7: Comparison with the method proposed by [TYC+20]

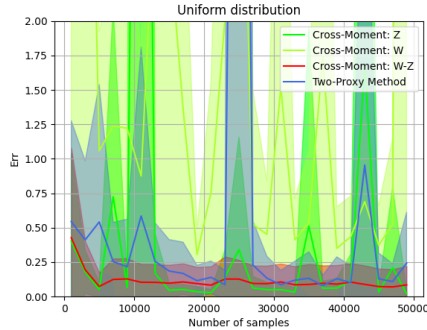

Figure 8: The performance measure $err$ against the number of samples. Colored regions represent the standard deviation of $err$.

method is more stable but "Cross-Moment W-Z" algorithm still performs better for high values of $\mathrm{Var}(\epsilon_w)/\mathrm{Var}(\epsilon_u)$.

### B.1.3   Experiments under model misspecification

In this scenario, we run experiments under model misspecification, i.e., the data generation mechanism does not follow the equations in (18). Instead, the data for the experiments is generated according to the following equations:

$$
\begin{aligned}
U &= \epsilon_u, \\
Z &= 10\tanh\left(\alpha_z U/a\right) + \epsilon_z, \\
D &= 10\tanh\left(\alpha_d U/a\right) + \epsilon_d, \\
Y &= \beta D + 10\tanh\left(\gamma U/a\right) + \epsilon_y,
\end{aligned}
\tag{19}
$$

where $a$ is some constant from the interval $[2, 10]$. All the exogenous noises are coming from the uniform distributions specified as in the previous experiments. Note that here we kept the linear relation from $D$ to $Y$ as it is challenging to quantify the causal effect with a single value if the relation is non-linear. In fact, in the non-linear case, the causal effect depends on the value of the treatment. For instance, one possible candidate to capture the causal effect is $\partial\mathbb{E}[Y|do(D := d)]/\partial d$, which is a function of $d$. The average relative error against the parameter $a$ is depicted in Figure 9. The relations from the latent confounder to the observed variables become more non-linear for lower values of $a$. Cross-Moment method still has a decent performance compared with the baselines.

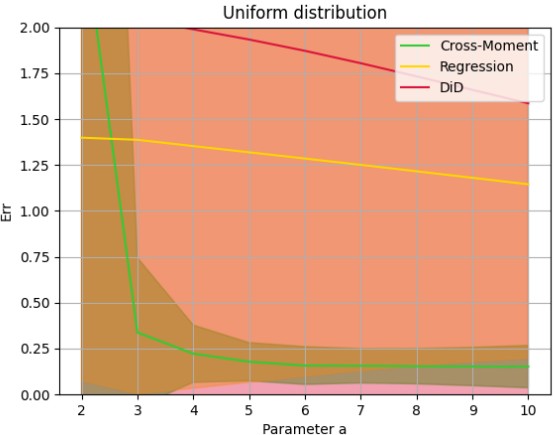

Figure 9: The performance measure $err$ against the parameter $a$ with the non-linear transformation $10\tanh(x/a)$. Colored regions represent the standard deviation of $err$.

## C  Derivations of DiD Estimator in Linear SCMs

Without loss of generality, we assumed that all the variables in the system are mean zero. Thus, there is no intercept term in the linear regression model: $\hat{Y} = \hat{\beta}_1 T + \hat{\beta}_2 D + \hat{\beta}DT$. Let $z_i$ be the outcome of the individual $i$ before assigning treatment. The mean of squared residual over the population before treatment is $\sum_i (z_i - \hat{\beta}_2 d_i)/n$ where $n$ is the size of the population and $d_i \in \{0, 1\}$ is equal to one if the treatment is assigned to individual $i$. Otherwise, $d_i$ is zero. For the post-treatment phase, let $y_i(d_i)$ be the outcome of individual $i$. Hence, the mean of squared residual over the population in the post-treatment phase is: $\sum_i (y_i(d_i) - \hat{\beta}_1 - (\hat{\beta}_2 + \hat{\beta})d_i)^2/n$. By performing a linear regression on the whole samples observed in pre-treatment and post-treatment phases over the population, we are minimizing the following risk: $\sum_i (z_i - \hat{\beta}_2 d_i)/n + \sum_i (y_i(d_i) - \hat{\beta}_1 - (\hat{\beta}_2 + \hat{\beta})d_i)^2/n$. Considering the uniform distribution among the individuals, the objective function in the minimization is equivalent to: $\min_{\hat{\beta}_1, \hat{\beta}_2, \hat{\beta}} \mathbb{E}[(Z - \hat{\beta}_2 D)^2] + \mathbb{E}[(Y - \hat{\beta}_1 - (\hat{\beta} + \hat{\beta}_2)D)^2]$. By taking partial derivative with respect to $\hat{\beta}_1$, $\hat{\beta}_2$, and $\hat{\beta}$ and setting them to zero, we can imply that $\hat{\beta}_1 = 0$, $\hat{\beta}_2 = \mathbb{E}[ZD]/\mathbb{E}[D^2]$, and $\hat{\beta} = (\mathbb{E}[YD] - \mathbb{E}[ZD])/\mathbb{E}[D^2]$, respectively. According to linear SCM in (3), we have $\mathbb{E}[YD] = \alpha_d(\alpha_d\beta + \gamma)\mathrm{Var}(\epsilon_u) + \beta\mathrm{Var}(\epsilon_d)$, $\mathbb{E}[ZD] = \alpha_d\alpha_z\mathrm{Var}(\epsilon_u)$, and $\mathbb{E}[D^2] = \alpha_d^2\mathrm{Var}(\epsilon_u) + \mathrm{Var}(\epsilon_d)$. By plugging these terms in the equation for $\hat{\beta}$, we get the equation in (7).

$$\begin{aligned}
U &= \epsilon_u \\
Z &= 10\tanh\left(\alpha_z U/a\right) + \epsilon_z \\
D &= 10\tanh\left(\alpha_d U/a\right) + \epsilon_d \\
Y &= \beta D + 10\tanh\left(\gamma U/a\right) + \epsilon_y
\end{aligned} \tag{20}$$