# OpenReview forum: "A Cross-Moment Approach for Causal Effect Estimation"
_NeurIPS.cc/2023/Conference — NeurIPS 2023 spotlight_

### Official Review · Reviewer_XfVt · 2023-06-26

**Soundness:** 4 excellent
**Presentation:** 4 excellent
**Contribution:** 3 good
**Rating:** 6
**Confidence:** 3

**Summary:**

This work proposes a cross-moment approach to estimating the average causal effect with latent confounders in linear SCM. One proxy variable of the latent confounder can be observed. In contrast to prior research (e.g., difference-in-difference) that requires stringent assumptions, this work shows that the causal effect can be identified and estimated using cross moments between the treatment, the outcome, and the proxy variable. It also discusses when the effect with latent confounder cannot be identified. Experiments on both synthetic and real-world data show its effectiveness.

**Strengths:**

1. This work introduces simple arithmetic operations on the cross moments to estimate causal effects with latent confounders in linear SCM. It addresses a conventional challenge in the field of solving an OICA problem and biased estimation of DiD, which may result in bad local optima.
2. It is technically sound and the idea is clearly and concisely described.
3. It can have significance in the field and practical importance given the prevalence of the studied problem.

**Weaknesses:**

1. The major concern is the evaluation. The baselines included are quite weak and old, e.g., KP14 published in 2014. Why the OICA method  [SGKZ20] has very poor performance is not clear. Other baselines may include proximal causal inference [1], for example. Also, for real-world data experiments, other baselines are not included. And the differences between the two methods when x is not included are not explained.
2. The limitations of the proposed approach are not discussed.
3. How applicable this method is is not clear.

[1] Mastouri, A., Zhu, Y., Gultchin, L., Korba, A., Silva, R., Kusner, M., ... & Muandet, K. (2021, July). Proximal causal learning with kernels: Two-stage estimation and moment restriction. In International Conference on Machine Learning (pp. 7512-7523). PMLR.

**Questions:**

1. Why SGKZ20 is very poor and what makes KP14 and the proposed approach much better than it?
2. What are the limitations of the proposed approach?
3. What are the potential applications of the proposed method?



I acknowledge I read the authors's response and I keep my positive score.

**Limitations:**

No limitations are discussed. Consider shortening Related Work and add limitations in the last section.

---

> ### Author Rebuttal · Authors · 2023-08-09
>
> **Regarding comments in weaknesses**
>
> 1. [Concerns about evaluations] Regarding the method in [SGKZ20], as mentioned in lines 70-71, it is based on solving an over-complete independent component analysis (OICA) which, in practice, can get stuck in bad local minima and return wrong results. In our experiments, we observed that it is often the case and it could not find the true mixing matrix and thus, has quite poor performance. We will add this explanation to the revised paper. The work mentioned by the reviewer requires at least two proxy variables $W,Z$ which may not be available in some settings. Moreover, they often need "Completeness" assumptions (such as in [MGTT18,TYC+20]), which in the discrete setting is equivalent to matrices $P(U|Z,d)$ and $P(W|U)$ be invertible. Nevertheless, we compared our algorithm with recent work [TYC+20] in proximal causal inference requiring two proxy variables, and the results are given in the attached file. The settings of experiments in Figure 2 and Figure 3 in the attached file are exactly the same as Figure 3(b) and Figure 5(a) in the submitted paper, respectively. As can be seen, Cross-Moment algorithm outperforms both methods in [KP14] and [TYC+20] when two proxy variables are available.
> 2. [About limitations] Please refer to the global response.
> 3. [Applicability] As mentioned in Introduction (lines 50-60), the causal graph we consider is applicable to settings in negative outcome control in the SCM framework as well as DiD in the potential outcome framework.  Moreover, our experiments in real dataset showcase our approach's effectiveness in the DiD setting.

---

> > ### Comment · Reviewer_XfVt · 2023-08-11
> >
> > I thank the authors for answering my questions and doubts. The authors have especially agreed to include the explanation for the bad performance of the method in [SGKZ20] in the next version and also presented addtional results to show their algorithm can outperform more recent models. I believe by including both will improve the paper, and hence I change my score to "Weak Accept". However, it is still not clear to me why the compared methods are different for synthetic data and real-world data.

---

> > > ### Author Response · Authors · 2023-08-11
> > >
> > > Thanks for reading the responses. We will add what was suggested by the reviewer in the revised paper. Regarding your question, please note that in the real dataset, we only have access to one proxy variable (employment level before the rise in the minimum wage). The methods in [KP14] or [TYC+20] require at least two proxy variables. Moreover, it is unclear which one of the covariates can be served as proxy variable W in these works.

---

### Official Review · Reviewer_5Myy · 2023-07-05

**Soundness:** 3 good
**Presentation:** 3 good
**Contribution:** 3 good
**Rating:** 6
**Confidence:** 4

**Summary:**

This work focuses on estimating the causal effect in the presence of an unmeasured confounder within a linear causal model. The authors demonstrate that the desired causal effect can be identified by employing a single proxy variable, leveraging its non-Gaussian characteristics. Additionally, they propose a Cross-Moment algorithm for estimating the model. Furthermore, they demonstrate the effectiveness of the proposed method through experiments on both synthetic and real-world datasets.


**Strengths:**

Estimating the causal effect becomes challenging in presence of unmeasured confounders.

The proposed sufficient identification condition of this paper is novel.

The analysis in this paper is presented in a logical manner.

This paper is clearly written.


**Weaknesses:**

The model is restricted as a linear causal model.

The sufficient identification condition is applicable only to a single unmeasured confounder, and the proposed method may not be suitable for cases involving multiple latent variables.


**Questions:**

Identification:

1.  ﻿Is the causal effect \beta identifiable under the Non-Gaussianity assumption? This is not clear to me. ﻿In my opinion, the reason why Assumptions 2 and 3 are introduced instead of non-gaussian assumption is because one need to use the cross-moment. Am I correct?
2. If Z directly affects D, the identification of the causal effect of D on Y may be affected. It would be helpful to investigate and discuss the potential implications of this scenario in the paper.
3. In the current setting, the paper assumes the existence of a single unmeasured confounder U. However, it is worth exploring and addressing the situation where there are multiple unmeasured confounders U. This could enhance the comprehensiveness and applicability of the proposed method.


Related work:

The following paper may be related to this work and is deserved to discuss.

Shuai, K., Luo, S., Zhang, Y., Xie, F., & He, Y. "Identification and Estimation of Causal Effects Using non-Gaussianity and Auxiliary Covariates." arXiv preprint arXiv:2304.14895 (2023).



**Limitations:**

Estimating the causal effect using the proposed method requires the availability of one proxy variable that is associated with the unmeasured confounder but does not directly affect the treatment variable. However, in practical scenarios, it can be challenging to identify or obtain such a proxy variable.

---

> ### Author Rebuttal · Authors · 2023-08-09
>
> **Regarding comments in weaknesses**
> 1. [Linearity assumption] Please refer to the global response.
> 2. [Single latent confounder] Please refer to the global response.
>
> **Questions**
> 1. As shown in Theorem 1, the causal effect is identifiable if condition in (5) is satisfied for the latent confounder. Moreover, as mentioned in lines 173-175, under some mild assumption (Assumption 3), this condition is equivalent to non-Gaussianity of latent confounder. Please note that the non-Gaussianity of latent confounder always implies the condition in (5). Moreover, in our results, we do not have any constraint on the type of distributions of the other observed variables in the system. Assumption 2 is indeed introduced so that all the cross moments are well-defined.
> 2. According to [SGKZ20], with non-Gaussian exogenous noises, if there is an edge from $Z$ to $D$ with the causal coefficient $\rho$, another model which is consistent with the same distribution over observed variables exists. The linear SCM equations for this model are as follows: $U:=\epsilon_z$, $Z:=U+\alpha_z\epsilon_u$, $D:=-(\alpha_d/\alpha_z)U+(\alpha_d/\alpha_z+\rho)Z+\epsilon_d$, $Y:=-(\gamma/\alpha_z)U+(\gamma/\alpha_z)Z+\beta D+\epsilon_y$. Incidentally, the causal effect in the second model is also $\beta$. Thus, without any prior knowledge about the causal structure, we can identify $\beta$ uniquely using the method in [SGKZ20] (which is based on OICA) as both models have the same causal effect from $D$ to $Y$. Regarding the Cross-Moment method, it can be shown that the same Eq. (4) holds for the causal graph with the additional edge from $Z$ to $D$. As a future work, it is interesting to devise a cross-moment method to identify $\alpha_d/\alpha_z$ in this setting.
> 3. We addressed this comment in the global response.
> 4. Thanks for mentioning it. We will discuss (Shuai et al. 2023) in the related work.
>
> **Regarding comments in limitations**
>
> As we discussed in Introduction (lines 50-60), the causal graph we consider is applicable to settings in negative outcome control in SCM framework as well as DiD in the potential outcome framework. Moreover, our experiments in real dataset showcase our approach's effectiveness in the DiD setting.

---

> > ### Comment · Reviewer_5Myy · 2023-08-12
> > **Regarding question 2**
> >
> > Thanks for your response! Regarding your example in Q2, what if we assume the faithfulness assumption, can we identify those two models from observed variables? In my view, the reason why the causal effect of $D$ on $Y$ can be uniquely identified is that the observed descendants of $U$ are not the same as the descendants of $D$. Please correct me if I'm wrong.

---

> > > ### Author Response · Authors · 2023-08-13
> > >
> > > Thanks for reading the responses. Under the faithfulness assumption, if the observational distribution is generated based on the original causal graph (with just an edge from $Z$ to $D$), the second model that we proposed in the response violates faithfulness assumption as $Z$ and $Y$ should be d-separated given $D$ and $U$ which is not the case in the causal graph of this model. Thus, under the faithfulness assumption, the original model is uniquely identifiable. Regarding uniquely recovering $\beta$, as mentioned by the reviewer, the latent confounder $U$ does not have the same observed descendants as $D$. Otherwise, we can swap their corresponding exogenous noises and get a new model with a different causal effect similar to the example in Section 4.1 of [SGKZ20].

---

> > > > ### Comment · Reviewer_5Myy · 2023-08-16
> > > > **Thanks for your response**
> > > >
> > > > Thanks for your response. The authors addressed my concern, so I will keep my score.

---

### Official Review · Reviewer_p1qb · 2023-07-07

**Soundness:** 3 good
**Presentation:** 3 good
**Contribution:** 3 good
**Rating:** 5
**Confidence:** 3

**Summary:**

This paper introduces an innovative technique for estimating the causal effect of a treatment on an outcome within linear structural causal models. This method utilizes cross moments, which are statistical moments derived from the joint distribution of the treatment and outcome variables, to quantify the causal effect. The authors demonstrate that this approach can relax the conventional assumption of a common trend in the difference-in-difference estimator, allowing for causal effect estimation in scenarios where traditional methods may fall short. To validate the effectiveness of the proposed method, the authors provide both simulation studies and a real-world application. These empirical analyses showcase the promising potential of this novel approach for estimating causal effects in linear structural causal models.

**Strengths:**

* Novelty: The paper introduces an innovative approach to estimating causal effects in linear structural causal models with latent confounders by leveraging cross moments. This method deviates from conventional approaches and exhibits the potential to yield more precise estimates within specific contexts.
* Rigor: The authors establish a rigorous theoretical framework for their proposed method, delineating the conditions that allow for the identification of the causal effect and the applicability of the method. Furthermore, they substantiate their claims through comprehensive simulation studies and a real-world application, bolstering the robustness and effectiveness of the approach.
* Significance: Estimating causal effects is a crucial task across various domains, and the proposed method holds substantial importance as it can potentially deliver more accurate estimates in specific scenarios. By addressing the limitations of traditional approaches, this method offers a valuable contribution to the field of causal effect estimation.


**Weaknesses:**

The authors should compare their proposed method to more existing methods for estimating causal effects, such as negative outcome control. This will help readers understand how the proposed method compares to existing methods in terms of accuracy and efficiency.


**Questions:**

See weakness

**Limitations:**

Discussed.

---

> ### Author Rebuttal · Authors · 2023-08-09
>
> **Regarding comments in weaknesses**
>
> As we mentioned in the global response, please note that most of these methods require at least two proxy variables and also further "Completeness" assumptions. Nevertheless, in the attachment of global response, we did additional experiments for comparing with more recent method in [TYC+20]. The settings of experiments in Figure 2 and Figures 3 in the attached file are exactly the same as Figure 3(b) and Figure 5(a) in the submitted paper, respectively. As can be seen, Cross-Moment algorithm outperforms both methods in [KP14] and [TYC+20] when two proxy variables exist.

---

> > ### Comment · Reviewer_p1qb · 2023-08-18
> > **Thanks for the reply**
> >
> >  The author successfully addressed my questions in their rebuttal stage, and I would like to keep my score to vote for acceptant.

---

### Official Review · Reviewer_dXHi · 2023-07-18

**Soundness:** 3 good
**Presentation:** 3 good
**Contribution:** 2 fair
**Rating:** 5
**Confidence:** 4

**Summary:**

The authors consider the estimation of causal effect in linear SCM with independent errors when there is a latent confounder U and one proxy variable of U (negative control outcome).

They generalize the DiD literature by relaxing the assumption of common trends and propose a general identification formula under the stated structural assumptions.

The authors show that under some restrictions on the latent confounder moments and on the U-Z, U-D relations, causal effects are uniquely identified nonparametrically using the identification formula. Furthermore, they propose a general estimation algorithm based on the cross-moments of Z and D.

In addition, the authors provide an ``impossibility result" which shows that under fully gaussian linear SCM, causal effects can not be uniquely identified.

They illustrate their proposed method in simulation and a data example.

**Strengths:**

Correcting the bias due to residual confounding using proxy variables (negative controls) is an emerging topic in causal inference. The authors consider the more difficult task that seeks to correct the bias with only one proxy variable.

Under the assumed linear structural model with independent errors, the authors utilized a well-known identity that can be thus used as an identification formula. Theorem 1, which provides the uniqueness guarantees, is novel and motivates a computationally easy estimation algorithm, which is an improvement in comparison to other proximal learning methods.

The formal results are rigorous and nontrivial. The theoretical guarantees provide a meaningful illustration of the limitations of the proposed identification formula.

The paper is well-written and easy to follow.



**Weaknesses:**

The authors provide results only for linear SCM with independent errors. Both linearity and exogenous errors are fairly strong assumptions that are not likely to hold in practice. Moreover, the theoretical results heavily depend on both assumptions and are not likely to extend to other SCMs.

On line 117, D is assumed to be a binary treatment. On line 119, the authors explicitly define the causal estimand of interest as the average causal effects on the treated. However, the SCM (line 133) states that $D = \alpha_dU +\varepsilon_d$, which, coupled with the assumption of independent zero mean errors (line 134), yields that
$$\Pr(D=1)=E[D]=\alpha_dE[U] + E[\varepsilon_d]=0$$
That is, if D is binary, the SCM implies that it is a deterministic random variable that equals 0 with probability one.
In addition, in Algorithm 1, $num$ is identically the same for all $n$ whenever $D$ is binary.
The authors are most likely well aware of this inconsistency since in the simulation study $D$ is not taken to be a binary variable.
Their proposed method works well for non-binary D, but causal estimands should be adjusted accordingly.

Section 3.2 (lines 209-228) is well known in the literature (see for example the recent review by Roth et al. 2023 ``What’s trending in difference-in-differences? A synthesis of the recent econometrics literature").

Experiments under misspecification (linearity, additional latent variables, etc.) are not presented. The robustness of the proposed methods is an open question.





**Questions:**

In the data example, estimation using the cross-moments algorithm is performed on the residuals of the outcom~covariates regression. Are there any theoretical guarantees (e.g., similar to Theorem 1) when covariates are included? are the covariates also need to have a linear relation to the treatment/outcome/negative control for the uniqueness of $\beta$?

Proximal learning (Tchegen Tchegen et al, cited by the authors) provides a flexible approach for estimating causal effects with latent confounders when there are at least two proxy variables. In practice, many studies do have more than one possible proxy variable. Do you think it is possible to extend the cross-moments algorithm for scenarios with more than one negative control (e.g., under linear SCM)?



**Limitations:**

As already stated, the theoretical results strongly rely on the linearity and independent error assumptions. The authors did not adequately address those limitations.

---

> ### Author Rebuttal · Authors · 2023-08-09
>
> **Regarding comments in weaknesses**
> 1. [About linearity and independent exogenous noises assumptions] Independence of exogenous errors is the main and standard assumption in structural causal models (SCM) which is based on the principle of independent mechanism (For more discussion on why this is a reasonable assumption, please see Section 2.1 in [1]). Regarding the linearity assumption, please refer to the global response.
>
> [1] Peters, Jonas, Dominik Janzing, and Bernhard Schölkopf. Elements of causal inference: foundations and learning algorithms. The MIT Press, 2017.
>
> 2. [About binary treatments] Section 2.1 (lines 117, 119) is referring to the DiD approach and it does not pertain to our assumption which we state later in our linear SCM setting (lines 133-134). As mentioned in line 134, without loss of generality, we can assume that all the variables have a mean equal to 0, as we can always achieve this by centering the data. It is noteworthy that our method is applicable to both discrete or continuous variables that satisfy the linear equations in (3). In the DiD setting, for instance, if there is a binary variable $D$ that takes values 0 or 1 with probability $1/2$ each, it can be represented in the SEM by a binary variable that equals $-1/2$ or $1/2$ with probability $1/2$ each. Please note that the experiments for the Minimum Wage and Employment dataset are done for the binary variable $D$, which shows that our method works well for the binary case. We will clarify these details in the revised paper.
> 3. [The reference "Roth et al. 2023"] Thanks for mentioning the reference. We will discuss it in the revised paper.
> 4. [Experiments under miss-specification] Regarding additional latent variables, please refer to the global response.
> About linearity assumption, we carried out some experiments under miss-specification in the linear relations. For the edges from the latent confounder $U$ to observed variables $Z,D,Y$, we replaced the linear relations with nonlinear functionalities. In particular, $Z= 10\tanh(\alpha_zU/a)+\epsilon_z$, $D= 10\tanh(\alpha_dU/a)+\epsilon_d$, and $Y= \beta D+ 10\tanh(\gamma U/a)+\epsilon_y$, where $a$ is some constant in $[2,10]$. Please note that we kept the linear relation from $D$ to $Y$ as it is challenging to quantify the causal effect with a single value if the relation is non-linear. In fact, in the non-linear case, the causal effect depends on the value of the treatment. For instance,  one possible candidate to capture the causal effect is $\partial \mathbb{E}[Y|do(D:=d)]/\partial d$,  which is a function of $d$.
> The average relative error against the parameter $a$  is depicted in Figure 1 in the attachment of the global response. The relations from the latent confounder to the observed variables become more non-linear for lower values of $a$. Cross-Moment method still has a decent performance compared with the baselines.
>
> **Questions**
> 1. In the case of covariates with linear relations, we can show that the causal effect can be identified uniquely if the latent confounder is not an ancestor of any covariates in the system. We will add a remark about this in the paper and also provide proof in the appendix. The main idea in the proof is to regress dependent variables on the covariates and reduce the problem to the one considered in the paper.
> 2. In our experiments (see lines 316-323), we extended our method to work with more than one proxy variable. As we mentioned above, the proposed method can be applied to linear SCMs with any number of latent confounders as long as for any latent confounder $U$, there exists a proxy variable such as $Z$, which is not an ancestor of $D$.

---

> > ### Comment · Reviewer_dXHi · 2023-08-16
> >
> > Thank you for the response. The authors addressed my questions and concerns.

---

### Author Rebuttal · Authors · 2023-08-09

# Global Response
We thank the reviewers for their time and valuable feedback. In the following, we provide a global response to some of the concerns/questions raised in the review.

**About linearity assumption:** The linearity assumptions present in a large body of the research in causal discovery/inference (for instance, see the survey in [1]) as it is important to see which learning tasks are feasible in this setting. Moreover, in practical scenarios where the available data is sparse, which is often the case in fields such as social science or medical research, linear models may serve as a good starting point. The use of more sophisticated models typically requires a larger dataset to train effectively. However, in situations where such extensive data is not available, the linear model can provide valuable insights, despite its simplicity. Therefore, while linear models may not capture the full complexity of real-world systems, they continue to be valuable tools in the domain of causal discovery and inference due to their interpretability and feasibility, especially when data is limited.

[1] Shimizu, Shohei. "LiNGAM: Non-Gaussian methods for estimating causal structures." Behaviormetrika 41 (2014): 65-98.

**About multiple latent confounders:** Regarding additional latent variables, the proposed method can be applied to linear SCMs with any number of latent confounders as long as for any latent confounder $U$, there exists a proxy variable $Z$, which is not an ancestor of $D$ (under a subtle change in Eq. (4)). The main idea is to adjust the values of $Cov(D,Y)$ and $Var(D)$ in Eq. (4) using proxies for each of corresponding latent variables similar to our treatment in the submitted version.
It is noteworthy that most previous work required at least two proxy variables $W,Z$, and even often came with some "Completeness" assumptions (such as in [MGTT18,TYC+20]), which in the discrete setting is equivalent to matrices $P(U|Z,d)$ and $P(W|U)$ be invertible.

We will add a remark about the above discussion in the revised paper. Please note that if a latent confounder does not come with any proxy variable, one can construct  SCMs in which the causal effect is not identifiable.

---

### Decision · Program_Chairs · 2023-09-21

**Decision:**

Accept (spotlight)

**Comment:**

The paper gives a strong theoretical result, giving a new way for estimating causal effects in linear SCMs with proxies for latent confounders, using only a single proxy, unlike previous approaches. This is an important advance and has applications for several widely used methods such as difference-in-differences and negative controls. While the result is restricted to linear SCMs, this genuinely novel result could open the door to useful follow up work in the field.